# Are foundation models for computer vision good conformal predictors?

**Leo Fillioux** *leo.fillioux@centralesupelec.fr*
*Université Paris-Saclay, CentraleSupélec, Gustave Roussy, INSERM, Cancer Data Science Unit, IHU PRISM National Precision Medicine Center in Oncology, Gif-sur-Yvette*

**Julio Silva-Rodríguez** *julio-jose.silva-rodriguez@etsmtl.ca*
*LIVIA, ILLS, ETS Montréal*

**Ismail Ben Ayed** *ismail.benayed@etsmtl.ca*
*LIVIA, ILLS, ETS Montréal*

**Paul-Henry Cournède** *paul-henry.cournede@centralesupelec.fr*
*Université Paris-Saclay, CentraleSupélec, Gustave Roussy, INSERM, Cancer Data Science Unit, IHU PRISM National Precision Medicine Center in Oncology, Gif-sur-Yvette*

**Maria Vakalopoulou** *maria.vakalopoulou@centralesupelec.fr*
*Université Paris-Saclay, CentraleSupélec, Gustave Roussy, INSERM, Cancer Data Science Unit, IHU PRISM National Precision Medicine Center in Oncology, Gif-sur-Yvette*

**Stergios Christodoulidis** *stergios.christodoulidis@centralesupelec.fr*
*Université Paris-Saclay, CentraleSupélec, Gustave Roussy, INSERM, Cancer Data Science Unit, IHU PRISM National Precision Medicine Center in Oncology, Gif-sur-Yvette*

**Jose Dolz** *jose.dolz@etsmtl.ca*
*LIVIA, ILLS, ETS Montréal*

**Reviewed on OpenReview:** *https://openreview.net/forum?id=Kxdg98gZp4*

## Abstract

Recent advances in self-supervision and contrastive learning have brought the performance of foundation models to unprecedented levels in a variety of tasks. Fueled by this progress, these models are becoming the prevailing approach for a wide array of real-world vision problems, including risk-sensitive and high-stakes applications. However, ensuring safe deployment in these scenarios requires a more comprehensive understanding of their uncertainty modeling capabilities, which has received little attention. In this work, we delve into the behaviour of vision and vision-language foundation models under Conformal Prediction (CP), a statistical framework that provides theoretical guarantees of marginal coverage of the true class. Across extensive experiments including popular vision classification benchmarks, well-known foundation vision models, and three CP methods, our findings reveal that foundation models are well-suited for conformalization procedures, particularly those integrating Vision Transformers. We also show that calibrating the confidence predictions of these models, a popular strategy to improve their uncertainty quantification, actually leads to efficiency degradation of the conformal set on adaptive CP methods. Furthermore, few-shot adaptation of Vision-Language Models (VLMs) to downstream tasks, whose popularity is surging, enhances conformal scores compared to zero-shot predictions. Last, our empirical study exposes APS as particularly promising in the context of vision foundation models, as it does not violate the marginal coverage guarantees across multiple challenging, yet realistic scenarios.

# 1    Introduction

Large-scale pre-trained vision foundation models, such as DINOv2 (Oquab et al., 2024), as well as those integrating text, such as CLIP (Radford et al., 2021), are driving a new learning paradigm in machine learning, achieving unprecedented results on a broad spectrum of tasks. Despite their desirable zero-shot and generalization capabilities, recent evidence has pointed out the existence of bias and factual errors in these models (Tu et al., 2023), which transcend the field of computer vision (Shuster et al., 2022; Barocas et al., 2023). For example, the original CLIP paper (Radford et al., 2021) demonstrated gender and race biases in certain zero-shot tasks, whereas Tu et al. (2023) identified that CLIP-based models are not always better calibrated than other arguably simpler ImageNet-trained models. Furthermore, Murugesan et al. (2024a) recently showed that adapted models magnify the miscalibration issue compared to the zero-shot setting, yielding overconfident predictions. These problems underscore widespread societal concerns surrounding the reliable deployment and use of foundation models in sensitive contexts, such as decision-making processes in critical scenarios, e.g., healthcare or security applications.

A popular solution to quantify the uncertainty present in the predictions of deep models is calibration. In this setting, the proposed strategies aim at reducing the discrepancies between model predictions and the actual correctness probability. Temperature Scaling (TS) (Guo et al., 2017), a simple variant of Platt Scaling (Platt et al., 1999), provides a simple post-processing approach to adjust the softmax probability scores of the trained models. Other lines of methods have proposed training objectives to enforce the model to produce less confident scores, either in the predictions space (Popordanoska et al., 2022; Pereyra et al., 2017), logits (Liu et al., 2023a; 2022), or modifying the ground truth labels (Mukhoti et al., 2020; Müller et al., 2019).

Conformal Prediction (CP) (Vovk et al., 1999), an alternative strategy to quantify the uncertainty (Sadinle et al., 2018; Romano et al., 2020; Angelopoulos et al., 2020), is a statistical framework which offers several advantages over calibration methods. First, unlike most methods in calibration, CP works directly on the model predictions, offering an appealing solution for black-box models. Second, instead of simply modeling the correctness of the predicted probability, CP produces a set of predictions, including the most likely classes, which can be of much interest in certain problems. Last, CP methods have theoretical guarantees for the marginal coverage of the true class within the predicted set, under several assumptions, contrary to calibration approaches.

Due to these properties, CP is gaining attention to conformalize the predictions of deep models (Barber et al., 2023; Angelopoulos et al., 2020; Ding et al., 2024; Sesia & Romano, 2021). Nevertheless, albeit the efforts to study CP in large language foundation models (Gui et al., 2024), its impact on vision foundation models has been unexplored, besides the significant implications it may have on a variety of vision problems. The aim of this study is to shed light and provide important insights into this direction. To achieve this, we conducted an extensive empirical analysis of the performance of three common CP methods on 17 popular vision foundation models across multiple vision datasets. Our extensive experiments further explore common situations encountered in practice, assessing the impact on CP: under distributional drifts, after confidence calibration and in few-shot adaptation to novel downstream tasks. Our key observations are:

(i) Vision, and vision-language foundation models seem to lead to lower set sizes and higher class-conditional coverage compared to their more traditionally (fully-supervised) trained counterparts.

(ii) Across all the experiments, Adaptive Prediction Sets (APS) is the best CP approach in terms of empirical coverage, while Regularized Adaptive Prediction Sets (RAPS) presents the best alternative from a conformal set size standpoint.

(iii) Under distributional shifts, APS exhibits the highest robustness among CP methods in terms of coverage guarantees, albeit decreasing its set efficiency.

(iv) Confidence calibration decreases the efficiency of conformal sets, but typically improves coverage gap

(v) Few-shot adaptation of vision-language models (VLMs) yields lower set sizes and coverage gaps than zero-shot predictions in ID data, with marginal gains on OOD.

(vi) Under domain shift, across different foundation models, those including visual transformers, such as DINO and CLIP, lead to smaller decreases in the conformal metrics compared to models integrating convolutional neural networks.

## 2 Related Work

**Foundation models for computer vision.** The landscape of foundation models has rapidly evolved in recent years. Traditionally, pre-trained convolutional networks based on ResNet architectures (He et al., 2016) were the main models used by the community. However, driven by the unprecedented advances in language models, e.g., GPT (Brown et al., 2020) or LLaMA (Touvron et al., 2023), as well as the vast availability of image data online, there are groundbreaking advances in unimodal (Caron et al., 2021; Oquab et al., 2024; Kirillov et al., 2023) and multimodal (Radford et al., 2021) foundation models for vision tasks, commonly based on vision transformers. These large pre-trained models aim to generalize across a broad span of visual tasks by pre-training on massive, diverse image datasets, exhibiting strong zero-shot and generalization capabilities to new tasks. For example, vision foundation models such as DINO (Caron et al., 2021; Oquab et al., 2024) rely on self-supervised learning strategies on large datasets, leading to excellent semantic understanding of visual content. On the other hand, CLIP (Radford et al., 2021) bridges the gap between language and vision modalities through contrastive learning, effectively allowing the model to understand images in the context of natural language prompts and enabling zero-shot capabilities.

**Quantifying the uncertainty of the predictions of deep networks** has recently garnered considerable interest. From a calibration standpoint, popular strategies include post-hoc approaches (Guo et al., 2017; Ding et al., 2021; Joy et al., 2023), which map the logits or softmax predictions to smoother distributions, and explicit *learning objectives* (Liu et al., 2022; 2023a; Müller et al., 2019; Mukhoti et al., 2020; Hebbalaguppe et al., 2022; Murugesan et al., 2024b), which are integrated into the loss function. Nevertheless, a main limitation of calibration methods is that they lack theoretical guarantees of model performance. In contrast, CP has recently emerged as a promising alternative, which provides marginal coverage guarantees over unseen test samples (Vovk et al., 2005; Shafer & Vovk, 2008). Specifically, CP resorts to a non-conformity score function (i.e., a measure of how "different" a particular data point is compared to a CP calibration dataset) to produce a finite prediction set, which is guaranteed to contain the true label with a user-specified confidence level. A central objective of the CP literature has been to improve either the *set efficiency* (i.e., smaller set sizes) or the class-conditional coverage. For this purpose, several non-conformity scores have been presented (Stutz et al., 2022; Romano et al., 2020; Angelopoulos et al., 2020; Sadinle et al., 2018; Einbinder et al., 2022; Ding et al., 2024; Straitouri et al., 2023), with Romano et al. (2020); Angelopoulos et al. (2020); Sadinle et al. (2018) being popular methods widely studied. A straightforward solution directly uses the raw class softmax predictions to generate the prediction sets (Sadinle et al., 2018). Adaptive Prediction Sets (APS) (Romano et al., 2020) provides an adaptive version, computing non-conformity scores by accumulating sorted softmax probabilities in descending order. To further improve the efficiency, RAPS (Angelopoulos et al., 2020) introduces an explicit regularization term, which penalizes non-conformity scores for unlikely classes.

However, a main limitation of existing evaluations is the focus on more traditional models, usually trained on data collection that falls within the calibration and test data points distribution. Despite this transfer learning framework not necessarily affecting the marginal guarantees provided by CP, how it affects its efficiency and conditional coverage remains to be explored. Thus, quantifying the uncertainty of their predictions is paramount given the rising popularity of foundation models in strategic domains. However, whereas uncertainty quantification from a calibration perspective has been scarcely studied (Murugesan et al., 2024a; Yoon et al., 2024; Tu et al., 2024), its exploration under CP is, to our knowledge, overlooked.

## 3 Background

### 3.1 Conformal Prediction Framework

Let $\mathcal{X}$ and $\mathcal{Y}$ denote the input and output space, respectively. We assume access to a calibration set $\mathcal{D}_{cal} = \{(\mathbf{x}_i, y_i)\}_{i=1}^n$ of $n$ independent and identically distributed (i.i.d.) samples, where each $\mathbf{x}_i = (p_{ik})_{1 \le k \le K}$ represents the black-box probabilities and $y_i \in \mathcal{Y} = \{1, 2, ..., K\}$ is the associated label. The goal of CP is to

construct a prediction set $\hat{C}(\mathbf{x}_{n+1}) \subseteq \mathcal{Y}$ for a new test input $\mathbf{x}_{n+1}$ such that it contains the true label $y_{n+1}$ with a user pre-specified coverage probability $1 - \alpha$, where $\alpha \in (0, 1)$ denotes the error level.

The core idea of CP is to assess the degree to which a new sample conforms to the underlying distribution of the calibration data by computing non-conformity scores. Let $S(\mathbf{x}, y)$ be a non-conformity measure (or scoring function) that assigns a score $s_i$ to each $(\mathbf{x}_i, y)$ pair, quantifying how unusual the pair is relative to the rest of the data. Given the non-conformity scores for all calibration examples and a new input $\mathbf{x}_{n+1}$ (unseen in the calibration set), the conformal prediction set $C(\mathbf{x}_{n+1})$ is defined as:

$$C(\mathbf{x}_{n+1}) = \{y \in \mathcal{Y} : S(\mathbf{x}_{n+1}, y) \leq q_\alpha\}, \tag{1}$$

where $q_\alpha$ is the $1 - \alpha$ quantile of the non-conformity scores on the calibration set, obtained with the observed labels:

$$q_\alpha = \text{QUANTILE}\left(\{S(\mathbf{x}_i, y_i)\}_{i=1}^N, \frac{\lceil (n+1)(1-\alpha) \rceil}{n}\right) \tag{2}$$

**Coverage Guarantees.** A key property of conformal prediction is its finite-sample *coverage guarantee*. This property ensures that the prediction sets achieve the desired coverage probability marginally over $\mathcal{X}$ and $\mathcal{Y}$, irrespective of the underlying data distribution, as long as the calibration and test data are exchangeable (Vovk et al., 2005). Formally, for any $1 - \alpha$, conformal predictors satisfy:

$$\mathbb{P}(y_{n+1} \in C(\mathbf{x}_{n+1})) \geq 1 - \alpha. \tag{3}$$

This property is crucial for applications requiring reliable uncertainty quantification, particularly where distributional assumptions (e.g., Gaussianity) may not hold.

**Tightness of Prediction Sets.** Conformal prediction guarantees valid marginal coverage. Nevertheless, the efficiency of the prediction sets, i.e., their size, depends on the choice of the non-conformity measure. Choosing an appropriate $S(\mathbf{x}, y)$ is key to balancing the trade-off between coverage and tightness of the prediction sets. In practice, we aim to minimize the size of the prediction sets while maintaining the desired coverage probability.

### 3.2 Non-conformity scores

**Least Ambiguous Classifier (LAC)** (Sadinle et al., 2018) aims to construct the smallest possible set under the assumption that the output is correct. Intuitively, it can be interpreted as a thresholding of the output probabilities for each category. Thus, the non-conformity score can be constructed as:

$$\mathcal{S}_{\text{LAC}}(\mathbf{x}, y) = 1 - \pi_x(y). \tag{4}$$

Where $\pi_x(y) = P(Y = y | X = x)$ is the softmax of the true class. LAC also provides notable efficiency in scenarios using an imperfect classifier. However, it lacks adaptability, e.g., in under-represented categories or uncertain predictions.

**Adaptive Prediction Sets (APS)** (Romano et al., 2020) provides a non-conformity score that leverages the accumulated confidence in the ordered probability predictions. Thus, APS is known to be an *adaptive score*, whose main objective is enhancing the coverage of uncertain predictions by sacrificing efficiency. Formally, APS is expressed as:

$$\mathcal{S}_{\text{APS}}(\mathbf{x}, y) = \rho(\mathbf{x}, y) + \pi_x(y) \cdot u, \tag{5}$$

where $\rho(\mathbf{x}, y)$ is the accumulated confidence of the categories more likely than the evaluated label $y$, i.e., $\rho(\mathbf{x}, y) = \sum_{k' \in \mathcal{Y}'(\mathbf{x}, y)} x_{k=k'}$, with $\mathcal{Y}'(\mathbf{x}, y) = \{k | x_k > x_{k=y}\}$. Adaptive methods usually include $u \in \{0, 1\}$, as a random variable to break ties to achieve exact marginal coverage.

**Regularized Adaptive Prediction Sets (RAPS)** (Angelopoulos et al., 2020) builds upon APS by adding a regularization term to enforce smaller predicted sets. Thus, APS score is modified to penalize the confidence of introducing additional, unlikely categories, after a certain set size is met:

$$\mathcal{S}_{\text{RAPS}}(\mathbf{x}, y) = \rho(\mathbf{x}, y) + \pi_x(y) \cdot u + \lambda(o(\mathbf{x}, y) - k_{reg})^+ \tag{6}$$

where $\lambda, k_{reg} \geq 0$ are hyper-parameters controlling the penalty strength, $o_x(y)$ is the rank of the sorted label, $o(\mathbf{x}, y) = |\mathcal{Y}'(\mathbf{x}, y)| + 1$, and $(\cdot)^+$ denotes the positive part.

We refer the reader to the different works (Angelopoulos et al., 2020; Sadinle et al., 2018; Romano et al., 2020) for the respective conformal calibration coverage guarantees.

## 4 Experiments

### 4.1 Experimental Setup

**Models:** We employ a total of 17 foundation models: two DINO (Caron et al., 2021) (DINO-S and DINO-B), four DINOv2 (Oquab et al., 2024) (DINOv2-S, DINOv2-B, DINOv2-L, and DINOv2-G), three VICReg (Bardes et al., 2022) (with ResNet-50, ResNet-50x2, and ResNet-200x2), and eight VLMs (five CLIP (Radford et al., 2021) models, MetaCLIP (Xu et al., 2024), LLaVa (Liu et al., 2023b), and Phi (Abdin et al., 2024)). Our main analysis is conducted on three popular vision datasets: CIFAR-10 (Krizhevsky & Hinton, 2009), CIFAR-100 (Krizhevsky & Hinton, 2009), and Imagenet (Deng et al., 2009), including its versions integrating domain shifts (Hendrycks et al., 2021a;b; Wang et al., 2019; Recht et al., 2019). Each dataset is split into two sets: one for training, and one for the conformal experiments. The latter is then split into one calibration set to tune the CP method, and one test set for evaluation. For few-shot, we adhere to the emerging CLIP few-shot literature (Zhang et al., 2022; Silva-Rodríguez et al., 2024; Zhou et al., 2022a), and evaluate models on 10 additional fine-grained and general concepts classification benchmarks: SUN397 (Xiao et al., 2010), FGVCAircraft (Maji et al., 2013), EuroSAT (Helber et al., 2018), StanfordCars (Krause et al., 2012), Food101 (Bossard et al., 2014), OxfordPets (Parkhi et al., 2012), Flowers102 (Nilsback & Zisserman, 2008), Caltech101 (Fei-Fei et al., 2004), DTD (Cimpoi et al., 2014), and UCF101 (Soomro et al., 2012).

**Metrics.** Given a pre-specified coverage probability $1 - \alpha$, a test set $\mathcal{D}_{\text{test}}$ and a method for predicting conformal sets $\hat{C}(\cdot)$, we resort to the following metrics.

$$\text{Set size}(\mathcal{D}_{\text{test}}) = \frac{1}{|\mathcal{D}_{\text{test}}|} \sum_{\mathbf{x} \in \mathcal{D}_{\text{test}}} |\hat{C}(\mathbf{x})| \tag{7}$$

$$\text{Cov}(\mathcal{D}_{\text{test}}) = \frac{1}{|\mathcal{D}_{\text{test}}|} \sum_{\mathbf{x}, y \in \mathcal{D}_{\text{test}}} \mathbb{1}_{y \in \hat{C}(\mathbf{x})} \tag{8}$$

$$\text{CovGap}(\mathcal{D}_{\text{test}}) = \frac{1}{|\mathcal{Y}|} \sum_{k \in \mathcal{Y}} \left| \text{Cov}(\mathcal{D}_{\text{test},k}) - (1 - \alpha) \right| \tag{9}$$

$$\text{MCCC}(D_{\text{test}}) = \min_{k \in \mathcal{Y}} \left( \text{Cov}(\mathcal{D}_{\text{test},k}) \right) \tag{10}$$

Where Equation 7 is the average predicted set size across the test set (sometimes referred to as efficiency), Equation 8 is the empirical marginal coverage, Equation 9 represents the average gap between the empirical coverage and the class-conditioned coverage, and Equation 10 is the min class-conditioned coverage (MCCC).

**Adaptation to target tasks.** The foundation models used in this study have been pre-trained using different strategies. Nevertheless, they need to be adapted for novel tasks, as their pre-trained versions do not accommodate classification tasks, i.e., there is no classification head. To do this, foundation models are frozen, and a linear probing (LP) head (one linear layer followed by a softmax activation function) is trained on each dataset by optimizing a cross-entropy loss (more details in Appendix Section A).

### 4.2 Results

To gain insights into the factors influencing the efficacy of CP in vision foundation models, we design four experiments. First, we explore the impact of CP in standard scenarios, where a large calibration set is

available to conform to the predictions of different models. Then, we challenge the status quo of CP and alter the conformal sets to accommodate real-world scenarios by including domain shifts. Furthermore, since confidence calibration is significantly linked to CP, we explore the impact of model calibration on CP performance. Last, we examine CP when adapting a very popular VLM, i.e., CLIP, to novel tasks.

### 4.2.1 Performance in the General Setting

First, we study the performance of 17 vision foundation models paired with CP methods under the standard setting, and on the three datasets, which present ideal conditions: a sufficiently large calibration set, and absence of distributional drifts between calibration and test sets. We aim to determine whether we can prescribe a winner solution in this scenario and which factors can help identify it.

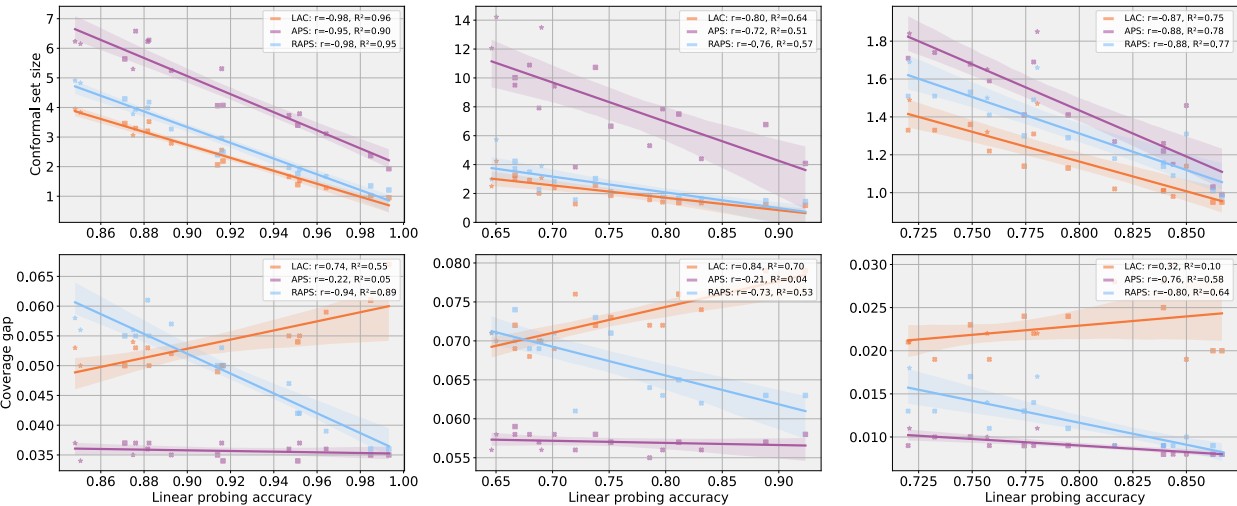

Figure 1: Relationship between the linear probing model accuracy and conformal set size (*top*) and the coverage gap (*bottom*) across different tasks of increasing complexity. From left to right: CIFAR-10, CIFAR-100 and ImageNet.

Figure 1 depicts the relationship between the linear probing performance of each model and the conformal set size (*top*), and coverage gap (*bottom*). The initial observation highlights a clear trend: higher-performance models tend to produce smaller prediction sets, regardless of the conformal method used. Nevertheless, while set efficiency (i.e. size) is typically considered as a sufficient condition in most prior literature in conformal prediction (Angelopoulos et al., 2020), our analysis revealed that *the relation with the accuracy does not consistently hold when examining other metrics*. In particular, Figure 1 (*bottom*) exposes that different CP methods yield mixed results for the coverage gap, which do not correlate directly with accuracy across all CP approaches. While APS appears to be almost unaffected by the model performance, RAPS clearly benefits from a strong performance, and LAC is negatively affected by more accurate models as the dataset becomes more complex.

Regarding comparison between the CP methods, if we consider the set size, APS is clearly outperformed by the other approaches, whereas LAC provides the smallest prediction sizes, closely followed by RAPS. Indeed, RAPS is specifically designed to reduce the conformal set size of APS. However, the coverage gap results indicate that this comes at the cost of increasing the range of the class-conditional coverage. Below, we analyze the underlying causes that may explain this behaviour.

*RAPS class-conditional coverage, and therefore coverage gap, are more sensitive to the model's accuracy.* Let us assume we have two models, $M_1$ and $M_2$, in a multi-class classification problem, whose accuracies are $Acc_1$ and $Acc_2$, respectively. For each class $y \in \mathcal{Y}$, we refer to $C_{M_i}(y) = \mathbb{P}(Y \in \mathcal{S}_{M_i}(X)|Y = y)$ as the class-conditional coverage for $y$ under model $M_i$, which measures the probability that predictions include the true label when the true label is $y$. Furthermore, let $\delta_{M_1} = \min_{y \in \mathcal{Y}} C_{M_1}(y)$ and $\delta_{M_2} = \min_{y \in \mathcal{Y}} C_{M_2}(y)$ denote the minimum class-conditional coverage achieved by each model. Under this scenario, we argue that

due to the penalty in RAPS, if $Acc_1 < Acc_2$, then $\delta_{M_1} < \delta_{M_2}$ when using RAPS as a conformal prediction method. In particular, $M_1$ (with lower accuracy) needs to expand its prediction sets for certain classes to meet the marginal coverage target $1-\alpha$. However, the penalty term encourages small prediction sets, limiting an excessive number of classes. Thus, for some difficult classes, model $M_1$ may still potentially fail to meet the target coverage, as the enforced penalty discourages overly large sets. This ultimately results in lower coverage rates for those specific classes than $M_2$. In contrast, since APS does not include any regularization term that encourages small set sizes, it will compensate for lower performing models by increasing its set sizes, ultimately attaining higher class-conditional coverages.

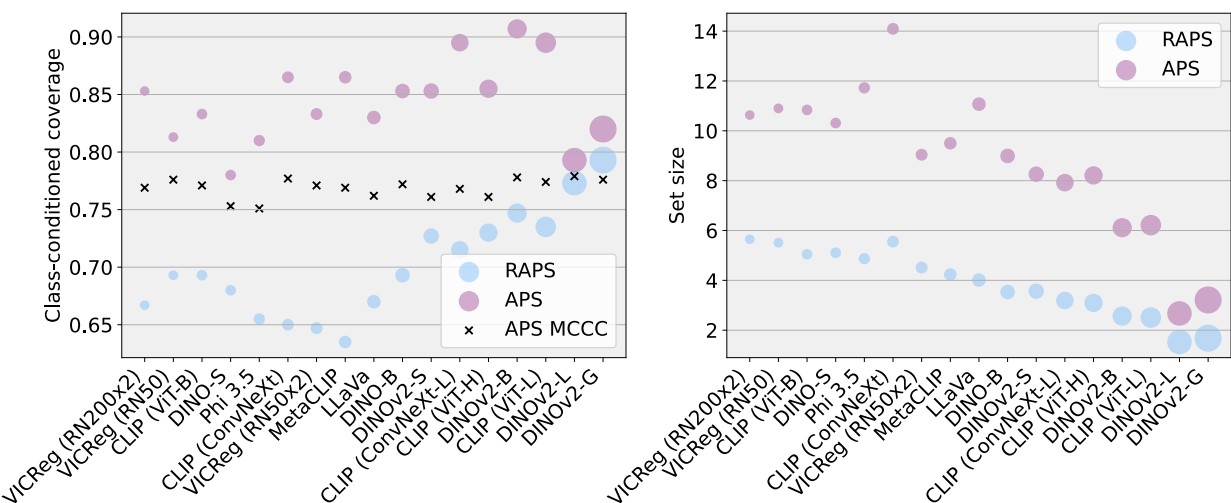

Figure 2: Comparison (APS *vs* RAPS) of the class-conditional coverage and set size for the class for which RAPS has the worst class-conditional coverage. Experiments performed on CIFAR-100. Models sorted (in ascending order) by their LP performance (min = 0.64 and max = 0.92), indicated by the size of the circles.

To confirm this hypothesis, we perform the following experiment. First, we identify the class with the lowest minimum class-conditional coverage obtained by RAPS for each model $m$, which we refer to as $y_m^R$, and find its corresponding set size, both represented blue circles in Figure 2. Then, we identify the class-conditional coverage and set size of the class $y_m^R$ provided by APS across all models, whose values are shown as pink circles in both plots of Figure 2. Last, we also include the minimum class-conditional coverage obtained by APS across each method, depicted with a cross. Note that the minimum class-conditional coverage from APS does not necessarily correspond to the coverage of class $y_m^R$. Upon close examination of these results, we observe that, indeed, for models presenting lower accuracy (those with smaller circles), the gap between the class-conditional coverage for the worst class in RAPS and the same class in APS is consistently larger than for more accurate models (larger circles). This empirical evidence shows that the minimum class-conditional coverage is substantially reduced because RAPS is constrained from expanding its predictions set size. This effect is more pronounced in less accurate models, where the true class may rank far away from the maximum allowable set size in the softmax predictions. In contrast, APS tolerates worse models by increasing the set size, which ultimately degrades the set efficiency but yields better class-conditional coverages.

Last, LAC presents structural differences with RAPS and APS, as it lacks an adaptive mechanism, relying on a uniform fixed threshold. Thus, LAC may yield inconsistent coverage rates across classes, resulting in high variability in the class-conditional coverage and thus in the coverage gap.

Following this analysis, we are also interested in determining whether a network pre-trained following a more traditional approach (i.e., standard supervised fine-tuning) offers similar conformal capabilities to self-supervised and contrastive ones. In particular, we select a ViT-B pre-trained on ImageNet, which is the same architecture as the visual encoder of the different foundation models. The results from this analysis (Table 1) reveal that, *despite obtaining lower classification accuracy when using LP on the different foundation models, CP methods typically yield better performance than in $ViT_{ImageNet}$*. This analysis should

Table 1: **SSL *vs* supervised learning**. Results on ImageNet obtained by CLIP (ViT-B), MetaCLIP (ViT-B), and DINO-S (ViT-S) and a ViT-B trained in a supervised manner on ImageNet.

| | Acc (↑) | Set size (↓) | | | MCCC (↑) | | | Coverage | | |
|---|---|---|---|---|---|---|---|---|---|---|
| | | LAC | APS | RAPS | LAC | APS | RAPS | LAC | APS | RAPS |
| $ViT_{CLIP}$ | 72.01 | 3.03 | 9.50 | 3.73 | 0.434 | **0.556** | 0.418 | 0.900 | 0.900 | 0.900 |
| $ViT_{DINO-S}$ | 74.92 | 3.27 | 10.02 | 4.23 | 0.433 | 0.477 | 0.412 | 0.900 | 0.900 | 0.900 |
| $ViT_{MetaCLIP}$ | 75.80 | 2.39 | **9.43** | **2.84** | **0.479** | 0.535 | **0.467** | 0.900 | 0.900 | 0.900 |
| $ViT_{ImageNet}$ | **76.08** | **2.36** | 38.75 | 4.46 | 0.416 | 0.495 | 0.405 | 0.900 | 0.900 | 0.900 |

be put in perspective with the analysis in Figure 1, which found that higher accuracy leads to lower set size, indicating that the training scheme plays a very important role. These differences are significant under the APS approach, where the set size is significantly degraded on $ViT_{ImageNet}$. Moreover, the class-conditional coverage is also substantially affected, with nearly 6% decrease compared to the best model. Note that our criterion for model selection in Table 1 was to ensure comparable predictive capabilities, essential to enable a fair assessment of their conformal performance, as differences in accuracy could otherwise confound the analysis. As DINO-B classification performance is not comparable, conclusions from a conformalization standpoint cannot be drawn from it (see Table 7).

To further delve into these differences, we compute, for each test sample, the difference between the conformal set size for APS when applied to $ViT_{ImageNet}$ and $ViT_{CLIP}$ models, whose distribution is depicted in Figure 3 (additional results in Appendix Section D). These values confirm that set size differences are not derived from a small set of isolated outliers but from a considerably large group of samples that see their conformal set increase when using the ViT trained in a supervised manner. These results suggest that the strategies used to train foundation models yield better CP metrics, resulting in conformalized models that can be deployed more safely on critical scenarios. It is important to stress that this study is limited due to the different dataset scales used for training (i.e., ImageNet alone is insufficient to train a foundation model). Our goal, however, is to understand the conformalization properties of readily available pre-trained models, regardless of how they were pre-trained.

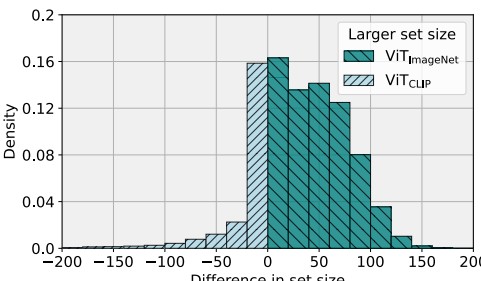

Figure 3: **$ViT_{ImageNet}$ *vs* $ViT_{CLIP}$.** Analyzing the difference in set size between a $ViT_{CLIP}$ and $ViT_{ImageNet}$. Equal set sizes not shown.

#### 4.2.2 Impact under Distribution Shifts

The theoretical guarantees of the coverage for conformal prediction hold under the hypothesis that the calibration set and the test set are drawn from the same distribution, i.e., *data exchangeability* assumption. In this section, we analyze the impact of having calibration sets that present distributional shifts with respect to the testing set.

We resort to ImageNet and its different versions: ImageNet-R (Hendrycks et al., 2021a), ImageNet-A (Hendrycks et al., 2021b), ImageNet-Sketch (Wang et al., 2019) and ImageNet-V2 (Recht et al., 2019). To introduce the distributional drift between the calibration and testing data, we adapt the pre-trained model to ImageNet. Then, ImageNet is used as the calibration set to conform the model, which is later tested on the ImageNet version used for adaptation. This is repeated for each ImageNet variant.

*APS* (Romano et al., 2020) *exhibits strong robustness against large distributional shifts, at the cost of substantially degrading efficiency.* One would expect that adaptive CP methods, such as APS and RAPS, somehow mitigate domain shifts due to their adaptive nature. Nevertheless, Figure 5 reveals several interesting obser-

vations, which contradict this intuition. First, we can observe that, when resorting to APS as CP method the coverage gap is consistently satisfied (or nearly satisfied) across all domain shifts and models (Figure 5, *middle*). In contrast, RAPS generally shows very similar performances compared to LAC, obtaining lower marginal coverage under several models and domains, and substantially lower than APS. To understand this phenomenon, we now study how set sizes evolve across domains for the different methods (Figure 5, *left*). We can easily observe that (*i*) APS yield the largest conformal sets across ImageNet domains, regardless of the model, and (*ii*) APS experiences the largest set increases when the complexity of the domain grows.

Thus, as exposed in the previous Section, APS satisfied marginal coverage by substantially including more predicted classes, therefore increasing conformal set sizes. The regularization term in RAPS, which pushes towards lower set sizes, limits the adaptation to the domain shift, coming at the cost of a decrease in coverage.

APS and RAPS are adaptive methods that produce similar minimum class-conditional coverage, as exhibited in ImageNet-A (Figure 5, *right*). However,

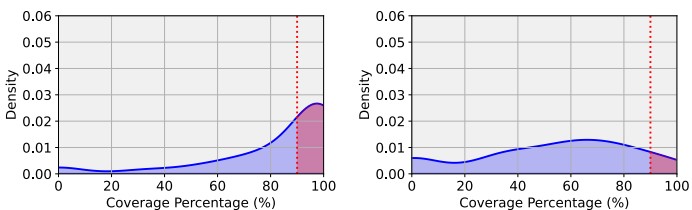

Figure 4: **Domain shift analysis.** Distribution of class-conditional coverages for CLIP (ViT-B) on ImageNet-A: APS (*left*) and RAPS (*right*).

they showcase significant differences in the average coverage gap. In the following, we explore this phenomenon in detail. More concretely, we depict in Figure 4 the distribution of the conditional class coverage values obtained by APS and RAPS on ImageNet-A for CLIP (ViT-B), more datasets and models in Appendix Section E. Interestingly, while both approaches see their minimum class-conditional coverage decrease, their distributions are completely different. Indeed, APS distributions exhibit a Gaussian shape, with a decreasing number of categories presenting lower conditional coverage as they separate from 1-$\alpha$. In contrast, the distribution in RAPS exposes a significantly worse scenario, where class-conditional coverages are almost uniformly spread, with a non-negligible amount of classes below the expected coverage, i.e., $1 - \alpha$.

Another interesting and valuable observation is that the low performance of all CP methods is magnified when coupled with VICReg (i.e., larger set sizes and lower coverage), suggesting that this family of foundation models may suffer more under distribution shifts.

### 4.2.3 Does calibration affect Conformal Prediction?

Confidence calibration is a popular strategy to improve the uncertainty estimates of deep models. These techniques, which can be either added as a post-processing step (Platt et al., 1999; Guo et al., 2017; Kull et al., 2019) or integrated as a training regularizer (Hebbalaguppe et al., 2022; Bohdal et al., 2023; Müller et al., 2020), adapt the model softmax predictions to reflect their actual performance accurately. The exponentially growing adoption of vision and vision-language foundation models in critical areas makes

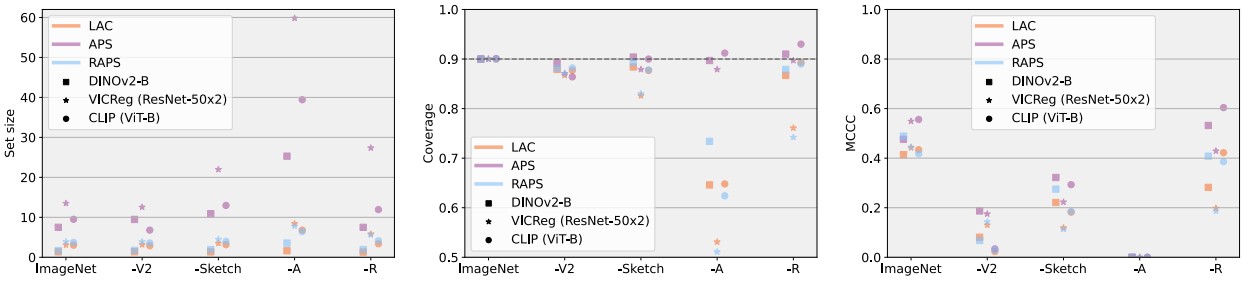

Figure 5: **Evaluation under domain-shift.** Set size ($\downarrow$), coverage ($\uparrow$), and MCCC ($\uparrow$) across three CP methods and three foundation models. ImageNet versions are sorted based on OOD performance in Silva-Rodríguez et al. (2024).

integrating confidence calibration a natural progression, as evidenced by recent works (Murugesan et al., 2024a; Yoon et al., 2024; Tu et al., 2024). Thus, in this section, we investigate this important issue, as the relationship between calibration and conformal prediction in vision foundation models remains largely unexplored. Specifically, we examine whether calibrating these models with the popular Temperature Scaling (TS) (Guo et al., 2017) affects the conformal performance of fixed and adaptive CP methods. Particularly, we apply TS to the ImageNet results in the general case (Section 4.2.1). As a proper validation set is not available, we evaluate the CP performance over a set of $T$ values (14 values from 0.85 to 2), and found that $T = 1.1$ typically yielded well-calibrated models[1]. In Appendix Section C, we explore histogram binning as another method for model calibration.

Our observations suggest that *calibration decreases the efficiency of CP sets, typically increasing the minimum class-conditional coverage, particularly on adaptive conformal prediction methods.*

**Effect on set efficiency.** Confidence calibration typically smooths the distribution of the class softmax scores, which results in less confident predictions. Consequently, the dominant value in the predicted softmax vector is typically lower in calibrated models. Nevertheless, since CP methods provide theoretical guarantees (under the *data exchangeability* assumption (Vovk et al., 2005)) to satisfy the target marginal coverage of $1 - \alpha$, these changes in the softmax distributions affect the conformalization obtained *pre-TS*. We present in Table 2 the results for the average set size and minimum class-conditional coverage before and after scaling the logits with TS. Furthermore, we include the Expected Calibration Error (ECE) values to verify that model calibration has improved.

These results show that if the model is calibrated, its prediction set size tends to be larger, particularly for adaptive CP approaches (RAPS, and more specifically, APS). Figure 6 further delves into these results, where we plot the distribution of differences between the set size of samples before and after calibration (i.e., a point in the distribution is $\mathcal{C}(x_i) - \mathcal{C}(x_i^{TS})$). We can identify that the overall larger size in APS is caused by a consistent efficiency degradation across samples and not a few atypical cases with large conformal sets.

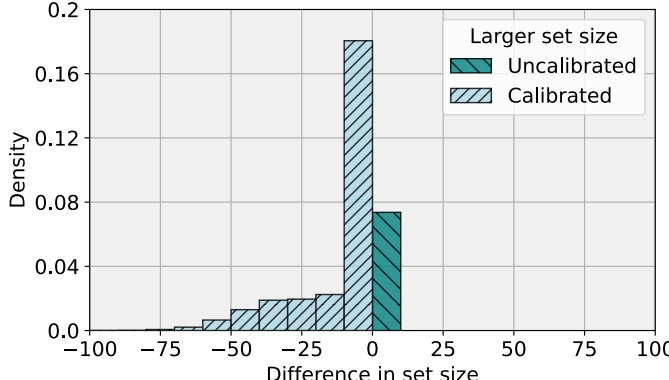

Figure 6: **Difference in conformal set size** (i.e., *efficiency*) when applying temperature scaling on APS and DINOv2-B ($T = 1.1$).

**Effect on class-conditional coverage.** We also observe that class-conditional coverage is typically improved on calibrated models, particularly when they are conformalized by adaptive CP methods. Thus, these results suggest that calibrating vision foundation models decreases efficiency while marginally enhancing class-conditional coverage, particularly on adaptive CP.

**Effect on coverage gap.** *The best coverage gap obtained by APS* (Romano et al., 2020) *approximately aligns with the coverage gap achieved at the optimal calibration point, whereas RAPS* (Angelopoulos et al., 2020) *coverage gap strongly differs.* Figure 7 shows the impact of the temperature $T$ on the different CP metrics on CLIP conformalized predictions. An interesting observation is that at the optimal temperature value $T = 1.1$, the CovGap of APS is very close to its optimal point[2] (0.0567 *vs* 0.0561). In contrast, while the RAPS coverage gap at optimal calibration is 0.0701, it decreases to 0.0642 as $T$ increases. This suggests that the coverage gap for APS tends to be associated with model calibration performance, with smaller gaps occurring near optimal calibration. On the downside, APS conformal sets efficiency is degraded. As $T$ increases, the softmax distributions get smoother, explaining the degradation in efficiency, which is even more drastic for APS, whose set size monotonically increases with $T$. We concede that, whereas APS

---

[1]Note that our goal is not to obtain the best-calibrated model but evaluate the impact of calibration on CP. Furthermore, this value aligns with existing works (Mukhoti et al., 2020; Joy et al., 2023) using TS for similar datasets, e.g., TinyImageNet.

[2]Results on Appendix Section C show similar behaviour for other models.

minimizes the coverage gap, it does so at the expense of increasing the conformal set size. However, we believe that increasing the size, up to some extent, while improving coverage gap is preferable in critical decision systems.

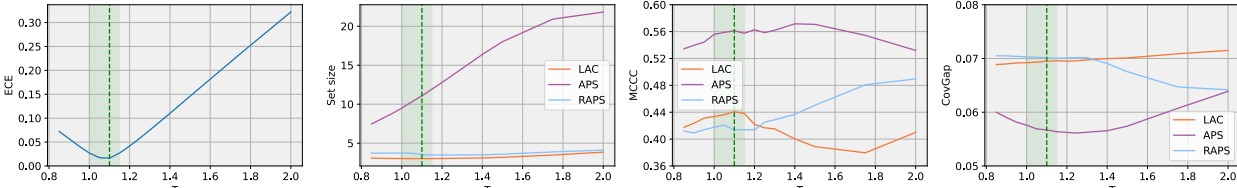

Figure 7: **Impact of the temperature** $T$ on ECE ($\downarrow$), set size ($\downarrow$), MCCC ($\uparrow$), and CovGap ($\downarrow$) (CLIP (ViT-B) on ImageNet). Green region indicates the area with ECE smaller than for $T = 1$. Green dotted line indicates value of optimal calibration ($T = 1.1$). More plots in Appendix Section C.

#### 4.2.4 Effect on few-shot adapted models

Adapting zero-shot CLIP models for downstream tasks in a few-shot labeled regime is becoming increasingly popular in VLMs. These strategies can be mainly categorized into Prompt Learning (Zhou et al., 2022b; Hantao Yao, 2023; Zhou et al., 2022a), which optimize the set of text prompts given to the text encoder, and Adapters (Yu et al., 2023; Lin et al., 2023; Huang et al., 2024; Silva-Rodríguez et al., 2024; Zhou et al., 2022b; Hantao Yao, 2023), where only a limited set of learnable parameters atop embedding representations is updated. Thus, questioning *whether adapting these models hinges the performance of CP methods* is of paramount importance, as it addresses a foundational aspect of effectively integrating uncertainty quantification with modern architectures. To assess the impact of CLIP adaptation in conformal prediction, we resort to representative methods of the few-shot adaptation families presented above, and follow standard adaptation and evaluation protocols in the literature (Silva-Rodríguez et al., 2024): (*i*) for out-of distribution (OOD), we adapt CLIP on few-shot samples ($M = 16$) from ImageNet, and evaluate on ImageNet and its variants, and (*ii*) for in distribution (ID) we adapt on few-shot samples and evaluate on the validation set from the same dataset.

*Few-shot VLM adaptation of pre-trained models lead to lower set sizes and coverage gaps* in the ID scenario (Table 3). First, we observe that, across both ID and OOD scenarios, both Adapters (ZSLP and CLAP) and Prompt Learning (CoOp and KgCoOp) yield smaller conformal set sizes and higher coverage gap than zero-shot CLIP, regardless of the CP method, with Prompt Learning yielding slightly better performances. In contrast, on the OOD scenario, only APS consistently enhances the set efficiency of the ZS model, with all the few-shot adaptation methods obtaining scarce coverage gap improvements across CP methods. These observations are related to the findings from the previous section, which suggested that better-calibrated models lead to larger conformal sets. Indeed, recent evidence (Murugesan et al., 2024a) demonstrated that

Table 2: **Quantitative impact of calibrating vision foundation models with TS.** Average size and minimum class-conditional coverage are reported between uncalibrated/calibrated models ($T = 1$ and $T = 1.1$ respectively) for several models, with ImageNet as benchmark dataset. Arrows indicate a decrease ($\blacktriangledown$) or an increase($\blacktriangle$) in metric after calibration, color indicates better (green) and worse (red) performance.

| | ECE ($\downarrow$) | Set size ($\downarrow$) | | | MCCC ($\uparrow$) | | |
|---|---|---|---|---|---|---|---|
| | ($\times 10^{-2}$) | LAC | APS | RAPS | LAC | APS | RAPS |
| DINOv2-S | 2.71/1.37▾ | 1.87/1.87 | 6.66/8.34▲ | 2.12/2.19▲ | 0.370/0.351▾ | 0.520/0.538▲ | 0.384/0.409▲ |
| DINOv2-B | 2.85/1.81▾ | 1.36/1.36 | 7.50/10.46▲ | 1.67/1.77▲ | 0.414/0.406▾ | 0.476/0.482▲ | 0.489/0.498▲ |
| DINOv2-L | 2.99/1.81▾ | 1.21/1.21 | 6.77/9.67▲ | 1.50/1.59▲ | 0.326/0.336▲ | 0.502/0.522▲ | 0.453/0.487▲ |
| DINOv2-G | 3.66/2.10▾ | 1.18/1.18 | 4.08/5.69▲ | 1.44/1.50▲ | 0.244/0.243▾ | 0.458/0.484▲ | 0.350/0.375▲ |
| VICReg (RN-50x2) | 2.34/2.16▾ | 3.08/3.09▲ | 13.49/16.06▲ | 3.89/3.80▾ | 0.442/0.430▾ | 0.549/0.557▲ | 0.445/0.446▲ |
| VICReg (RN-100x2) | 2.21/1.90▾ | 2.50/2.49▾ | 12.05/14.61▲ | 2.98/3.01▲ | 0.440/0.430▾ | 0.567/0.566▾ | 0.462/0.471▲ |
| CLIP (ViT-B) | 2.70/1.63▾ | 3.03/3.01▾ | 9.50/11.11▲ | 3.73/3.53▾ | 0.434/0.441▲ | 0.556/0.564▲ | 0.418/0.414▾ |
| MetaCLIP | 2.63/1.99▾ | 2.39/2.40▲ | 9.43/11.31▲ | 2.84/2.84 | 0.479/0.477▾ | 0.535/0.541▲ | 0.467/0.469▲ |

Table 3: **Few-shot adaptation (16-shots).** Set size and CovGap for CLIP (ViT-B backbone) on *in-distribution* (average over 11 datasets) and *out-of-distribution* (average over ImageNet versions). Additional results are presented in Appendix Section F.

| | | Set size (↓) | | | CovGap (↓) | | |
|---|---|---|---|---|---|---|---|
| | | LAC | APS | RAPS | LAC | APS | RAPS |
| **ID** | ZS | 5.29 | 6.98 | 6.49 | 0.114 | 0.094 | 0.102 |
| | ZSLP | 2.09 | 3.43 | 2.53 | 0.087 | 0.060 | **0.064** |
| | CLAP (Silva-Rodríguez et al., 2024) | 2.13 | 3.52 | 2.57 | 0.088 | 0.060 | 0.065 |
| | CoOp (Zhou et al., 2022b) | **2.07** | **2.87** | **2.47** | **0.083** | 0.059 | 0.067 |
| | KgCoOp (Hantao Yao, 2023) | 2.10 | 3.11 | 2.53 | 0.085 | **0.059** | 0.065 |
| **OOD** | ZS | 7.68 | 19.22 | 9.93 | 0.095 | 0.085 | 0.094 |
| | ZSLP | 8.34 | 18.51 | 10.61 | **0.092** | **0.083** | 0.092 |
| | CLAP (Silva-Rodríguez et al., 2024) | 7.55 | 17.41 | 9.89 | 0.093 | **0.083** | **0.091** |
| | CoOp (Zhou et al., 2022b) | 7.86 | **16.35** | 10.03 | 0.093 | 0.085 | 0.095 |
| | KgCoOp (Hantao Yao, 2023) | **7.54** | 16.68 | **9.79** | 0.094 | 0.085 | 0.095 |

few-shot CLIP adaptation methods deteriorate the confidence estimates compared to ZS predictions, which, following our observations, should result in smaller conformal sets, as validated in Table 3. Further details are provided in Appendix Section F.

## 5 Conclusion

In this study, we aimed at answering the question: *Which CP method and model should I use, and what can I expect, in the era of vision foundation models*? Our findings revealed that vision foundation models yield better conformal metrics than their traditional pre-trained counterparts, with models integrating visual transformers outperforming those that use convolutional neural networks, particularly under domain shifts. Furthermore, we observed that several common situations encountered in practice (i.e., presence of distributional drifts and models undergoing confidence calibration) are indeed detrimental for some CP approaches. Interestingly, an adaptive CP method, i.e., APS, exhibited stronger robustness to these scenarios, particularly in terms of conditional coverage, but at the expense of degrading the set efficiency.

The final decision ultimately hinges on the requirements of each task. In certain fields, e.g., medical diagnosis, it may be preferable to maximize the conditional coverage across classes, even if it increases the conformal set size, as this minimizes the risk of critical errors. In contrast, in domains where errors are less consequential, maintaining smaller set sizes might be preferable to streamline decision-making and reduce computational demands. Thus, the choice depends on balancing the trade-offs between accuracy, usability, and the specific demands of the application.

### Acknowledgement

This work has benefited from state financial aid, managed by the Agence Nationale de Recherche under the investment program integrated into France 2030, project reference ANR-21-RHUS-0003. This work was partially supported by the ANR Hagnodice ANR-21-CE45-0007. This work was partially supported by the ANR DAFNI ANR-23-CE45-0029. This work was granted access to the HPC resources of IDRIS under the allocation 2023-AD011014802 made by GENCI. We gratefully acknowledge the DATAIA program for supporting JD as a visiting professor at Université Paris-Saclay. This work was partially supported by IA CLUSTER program, reference ANR-23-IACL-0003 – DATAIA CLUSTER.

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
