## A    Models Used and Implementation Details

**Foundation models**. We use 17 foundation models for vision tasks from three categories. They differ in the number of modalities for training (i.e., vision *vs* vision-language), strategy used for training (i.e., contrastive *vs* self-supervised learning), and backbones used (i.e., vision transformers *vs* convolutional neural networks). More conctretely, the DINO (Caron et al., 2021; Oquab et al., 2024) models employed in this work are ViT-based models trained in a self-supervised manner. CLIP (Radford et al., 2021) is a vision-language model, which trains a vision and a text encoder in a contrastive fashion. Last, VICReg (Bardes et al., 2022) are CNNs trained in a self-supervised manner using contrastive learning. Table 4 summarizes the models used for this study.

**Implementation details.** All linear probing heads are trained with a cross-entropy loss with the Adam (Kingma & Ba, 2017) optimizer and a learning rate of $10^{-4}$, whose patience is set to 10. The implementation of the training is based on PyTorch (Paszke et al., 2019), and the conformal predictions methods on TorchCP (Wei & Huang, 2024). For RAPS, we used $k_{\text{RAPS}} = 2$ and $\lambda_{\text{RAPS}} = 0.1$ as hyperparameters. Across all experiments, we use $\alpha = 0.1$, except for CIFAR-10 for which we use $\alpha = 0.05$.

## B    Relationship Between Model Performance and CP Metrics

Table 5, Table 6, and Table 7 report the numerical values corresponding to Figure 1, showcasing the relationship between the linear probing accuracy of each model, and the CP performance in terms of set size and coverage gap for CIFAR-10, and CIFAR-100, and ImageNet respectively.

## C    Model Calibration

### C.1    Temperature Scaling

We report in Table 8 a more complete version of the results presented in Table 2, containing all explored models.

Additionally, Figures 10, 11, 12, 13, 14, 15, 16, 17, 18 showcase the impact of temperature $T$ on the ECE, set size, minimum class-conditional coverage, and coverage gap across all foundation models. The observations on these figures *strongly align with the findings on the main paper as, regardless of the model, APS yields optimal coverage gap close to the optimal temperature point.* Indeed, looking at the different models (i.e., DINO-based, VICReg and CLIP), we can observe that the behaviour of RAPS (in terms of coverage gap), strongly varies across models. Additionally, Figure 8 shows the evolution of $q_\alpha$, the threshold on the conformal scores $s$ before and after calibration. When applying temperature scaling, the distribution of softmax across classes approaches a uniform distribution, lowering the scores for the most likely classes. Intuitively, this means that more classes will need to be included in the set to ensure coverage, leading to a decrease in threhsold $q_\alpha$. This decrease is consistent with the observed increase in set size, which can be seen in Fig. 9, showing the difference in set size when applying temperature scaling for APS on CLIP. Note that this trend holds across models, as a similar behaviour for DINO is observed in Fig. 6 in the main paper. Note that both Fig. 6 and 9 only show cases where the set sizes are different between the calibrated and uncalibrated model.

### C.2    Histogram Binning

In Table 9, we explore histogram binning as an alternative method (other than temperature scaling) for model calibration. This method sorts predictions into a predefined number of bins (100 in our case) and adjusts the predictions for the confidence score to match the correct frequency. This is learned on a separate set and applied on a test set. In particular, with this approach as calibration technique we observed an increase in set size, similar to TS, with a slight decrease in MCCC, *aligning with our observations on the effects of calibration on conformalization.*

## D  Conformal set size analysis

We show the difference between the conformal set size for APS when applied to $\text{ViT}_{\text{ImageNet}}$ and $\text{ViT}_{\text{DINO-S}}$ and for $\text{ViT}_{\text{ImageNet}}$ and $\text{ViT}_{\text{MetaCLIP}}$ in Figure 19, similarly to what is shown in Figure 5.

## E  Domain Shift

Table 10 reports the numerical values for Fig. 5, showcasing APS's strong performance in terms of marginal and conditional coverage under distribution shift, at the cost of decreasing the set efficiency. Figure 20 depict the distribution of class-conditional coverage for DINOv2-B, VICReg (RN 50x2), and CLIP (ViT-B). The kernel density estimate plots show the class-conditional coverage for a distribution shift with ImageNet-A (*top*), ImageNet-R (*middle*), and ImageNet-V2 (*bottom*), for APS (*left*) and RAPS (*right*). These results clearly demonstrate the stronger resistance to distribution shift for APS compared to RAPS, even when both methods show a minimum class-conditional coverage of 0.

## F  Few-Shot Adaptation

We present in Table 11 and Table 13 a detailed version of Table 3 for 16 shots, where the *OOD* section (in Table 3) corresponds to the average across all ImageNet variants of Table 11, and the *ID* section corresponds to the average across all 11 datasets of Table 13. Table 12 shows in distribution results for adapting only 4 shots, where we observe slightly higher set size and coverage gap compare to the results for 16 shots.

Table 4: **Summary of the models used.** Models used along with the number of parameters, training scheme, modalities used, and architecture type.

|  | Num. of parameters | Training scheme | Modalities | Architecture |
|---|---|---|---|---|
| DINO-S | 21,665,664 | SSL | Vision | ViT |
| DINO-B | 85,798,656 | SSL | Vision | ViT |
| DINOv2-S | 22,056,576 | SSL | Vision | ViT |
| DINOv2-B | 86,580,480 | SSL | Vision | ViT |
| DINOv2-L | 304,368,640 | SSL | Vision | ViT |
| DINOv2-G | 1,136,480,768 | SSL | Vision | ViT |
| VICReg (ResNet-50) | 23,508,032 | SSL | Vision | CNN |
| VICReg (ResNet-50x2) | 93,907,072 | SSL | Vision | CNN |
| VICReg (ResNet-200x2) | 250,128,512 | SSL | Vision | CNN |
| MetaCLIP | 149,620,737 | Contrastive | Vision-Language | ViT |
| Phi 3.5 | 303,507,456 | SSL+Supervised | Vision-Language | ViT |
| LLaVa | 303,507,456 | SSL+Supervised | Vision-Language | ViT |
| CLIP (ViT-B) | 87,456,000 | Contrastive | Vision-Language | ViT |
| CLIP (ViT-L) | 303,966,208 | Contrastive | Vision-Language | ViT |
| CLIP (ViT-H) | 632,076,800 | Contrastive | Vision-Language | ViT |
| CLIP (ConvNeXt) | 88,221,824 | Contrastive | Vision-Language | CNN |
| CLIP (ConvNeXt-L) | 199,770,816 | Contrastive | Vision-Language | CNN |

Table 5: **Linear probing performance and conformal metrics for CIFAR-10.** Linear probing F1 score, and corresponding set size and MCCC for LAC, APS, and RAPS ($\alpha = 0.05$).

| | **F1** ($\uparrow$) | **Set size** ($\downarrow$) | | | **MCCC** ($\uparrow$) | | |
|---|---|---|---|---|---|---|---|
| | | LAC | APS | RAPS | LAC | APS | RAPS |
| DINO-S | 0.8711 | 1.36 ± 0.00 | 1.68 ± 0.00 | 1.53 ± 0.00 | 0.885 ± 0.001 | 0.930 ± 0.001 | 0.913 ± 0.001 |
| DINO-B | 0.9165 | 1.14 ± 0.00 | 1.41 ± 0.00 | 1.30 ± 0.00 | 0.896 ± 0.001 | 0.932 ± 0.001 | 0.921 ± 0.001 |
| DINOv2-S | 0.9140 | 1.13 ± 0.00 | 1.41 ± 0.00 | 1.29 ± 0.00 | 0.899 ± 0.001 | 0.932 ± 0.001 | 0.927 ± 0.001 |
| DINOv2-B | 0.9511 | 1.01 ± 0.00 | 1.22 ± 0.00 | 1.14 ± 0.00 | 0.899 ± 0.001 | 0.934 ± 0.001 | 0.933 ± 0.001 |
| DINOv2-L | 0.9848 | 0.95 ± 0.00 | 1.03 ± 0.00 | 1.01 ± 0.00 | 0.893 ± 0.001 | 0.933 ± 0.001 | 0.933 ± 0.001 |
| DINOv2-G | 0.9932 | 0.95 ± 0.00 | 0.99 ± 0.00 | 0.98 ± 0.00 | 0.910 ± 0.001 | 0.936 ± 0.001 | 0.934 ± 0.001 |
| VICReg (ResNet-50) | 0.8506 | 1.49 ± 0.00 | 1.84 ± 0.00 | 1.69 ± 0.00 | 0.902 ± 0.001 | 0.925 ± 0.001 | 0.905 ± 0.001 |
| VICReg (ResNet-50x2) | 0.8749 | 1.32 ± 0.00 | 1.65 ± 0.00 | 1.50 ± 0.00 | 0.910 ± 0.001 | 0.930 ± 0.001 | 0.923 ± 0.001 |
| VICReg (ResNet-200x2) | 0.8481 | 1.47 ± 0.00 | 1.85 ± 0.00 | 1.66 ± 0.00 | 0.907 ± 0.001 | 0.925 ± 0.001 | 0.912 ± 0.001 |
| MetaCLIP | 0.8926 | 1.22 ± 0.00 | 1.59 ± 0.00 | 1.41 ± 0.00 | 0.903 ± 0.001 | 0.933 ± 0.001 | 0.928 ± 0.001 |
| Phi 3.5 | 0.8761 | 1.33 ± 0.00 | 1.74 ± 0.00 | 1.51 ± 0.00 | 0.906 ± 0.001 | 0.928 ± 0.001 | 0.921 ± 0.001 |
| LLaVa | 0.9159 | 1.14 ± 0.00 | 1.46 ± 0.00 | 1.31 ± 0.00 | 0.902 ± 0.001 | 0.935 ± 0.001 | 0.928 ± 0.001 |
| CLIP (ViT-B) | 0.8823 | 1.33 ± 0.00 | 1.71 ± 0.00 | 1.51 ± 0.00 | 0.909 ± 0.001 | 0.931 ± 0.001 | 0.921 ± 0.001 |
| CLIP (ViT-L) | 0.9642 | 0.98 ± 0.00 | 1.15 ± 0.00 | 1.09 ± 0.00 | 0.877 ± 0.001 | 0.934 ± 0.001 | 0.932 ± 0.001 |
| CLIP (ViT-H) | 0.9519 | 1.01 ± 0.00 | 1.26 ± 0.00 | 1.16 ± 0.00 | 0.882 ± 0.001 | 0.933 ± 0.001 | 0.932 ± 0.001 |
| CLIP (ConvNeXt) | 0.8817 | 1.31 ± 0.00 | 1.69 ± 0.00 | 1.49 ± 0.00 | 0.904 ± 0.001 | 0.933 ± 0.001 | 0.922 ± 0.001 |
| CLIP (ConvNeXt-L) | 0.9470 | 1.02 ± 0.00 | 1.27 ± 0.00 | 1.18 ± 0.00 | 0.876 ± 0.001 | 0.932 ± 0.001 | 0.932 ± 0.001 |

Table 6: **Linear probing performance and conformal metrics for CIFAR-100.** Linear probing F1 score, and corresponding set size and MCCC for LAC, APS, and RAPS ($\alpha = 0.1$).

| | **F1** ($\uparrow$) | **Set size** ($\downarrow$) | | | **MCCC** ($\uparrow$) | | |
|---|---|---|---|---|---|---|---|
| | | LAC | APS | RAPS | LAC | APS | RAPS |
| DINO-S | 0.6668 | 3.46 ± 0.01 | 5.64 ± 0.01 | 4.29 ± 0.02 | 0.678 ± 0.003 | 0.753 ± 0.003 | 0.682 ± 0.003 |
| DINO-B | 0.7379 | 2.19 ± 0.01 | 4.07 ± 0.01 | 2.50 ± 0.01 | 0.667 ± 0.003 | 0.772 ± 0.003 | 0.678 ± 0.003 |
| DINOv2-S | 0.7515 | 2.06 ± 0.01 | 4.06 ± 0.01 | 2.40 ± 0.02 | 0.689 ± 0.004 | 0.761 ± 0.004 | 0.700 ± 0.004 |
| DINOv2-B | 0.8116 | 1.39 ± 0.01 | 3.40 ± 0.01 | 1.76 ± 0.00 | 0.671 ± 0.003 | 0.778 ± 0.003 | 0.739 ± 0.003 |
| DINOv2-L | 0.8885 | 1.02 ± 0.01 | 2.37 ± 0.01 | 1.35 ± 0.00 | 0.661 ± 0.003 | 0.779 ± 0.003 | 0.771 ± 0.002 |
| DINOv2-G | 0.9235 | 0.95 ± 0.00 | 1.93 ± 0.01 | 1.21 ± 0.00 | 0.583 ± 0.004 | 0.776 ± 0.003 | 0.762 ± 0.003 |
| VICReg (ResNet-50) | 0.6503 | 3.83 ± 0.01 | 6.15 ± 0.01 | 4.83 ± 0.02 | 0.696 ± 0.004 | 0.776 ± 0.002 | 0.702 ± 0.003 |
| VICReg (ResNet-50x2) | 0.6902 | 3.06 ± 0.01 | 5.30 ± 0.01 | 3.78 ± 0.01 | 0.670 ± 0.004 | 0.771 ± 0.002 | 0.658 ± 0.004 |
| VICReg (ResNet-200x2) | 0.6461 | 3.95 ± 0.01 | 6.23 ± 0.02 | 4.91 ± 0.01 | 0.713 ± 0.003 | 0.769 ± 0.002 | 0.687 ± 0.003 |
| MetaCLIP | 0.7015 | 2.79 ± 0.01 | 5.25 ± 0.01 | 3.27 ± 0.01 | 0.683 ± 0.004 | 0.769 ± 0.003 | 0.652 ± 0.003 |
| Phi 3.5 | 0.6796 | 3.30 ± 0.01 | 6.58 ± 0.02 | 3.94 ± 0.01 | 0.650 ± 0.004 | 0.751 ± 0.003 | 0.654 ± 0.003 |
| LLaVa | 0.7201 | 2.56 ± 0.01 | 5.31 ± 0.02 | 2.97 ± 0.01 | 0.701 ± 0.004 | 0.762 ± 0.003 | 0.694 ± 0.003 |
| CLIP (ViT-B) | 0.6667 | 3.52 ± 0.01 | 6.28 ± 0.02 | 4.18 ± 0.01 | 0.705 ± 0.003 | 0.771 ± 0.002 | 0.691 ± 0.003 |
| CLIP (ViT-L) | 0.8315 | 1.28 ± 0.00 | 3.11 ± 0.01 | 1.67 ± 0.00 | 0.644 ± 0.003 | 0.774 ± 0.003 | 0.738 ± 0.003 |
| CLIP (ViT-H) | 0.7973 | 1.53 ± 0.00 | 3.79 ± 0.01 | 1.91 ± 0.00 | 0.653 ± 0.004 | 0.761 ± 0.003 | 0.709 ± 0.004 |
| CLIP (ConvNeXt) | 0.6880 | 3.21 ± 0.00 | 6.23 ± 0.02 | 3.99 ± 0.01 | 0.692 ± 0.003 | 0.777 ± 0.003 | 0.648 ± 0.004 |
| CLIP (ConvNeXt-L) | 0.7858 | 1.66 ± 0.00 | 3.73 ± 0.01 | 1.97 ± 0.00 | 0.680 ± 0.004 | 0.768 ± 0.002 | 0.714 ± 0.003 |

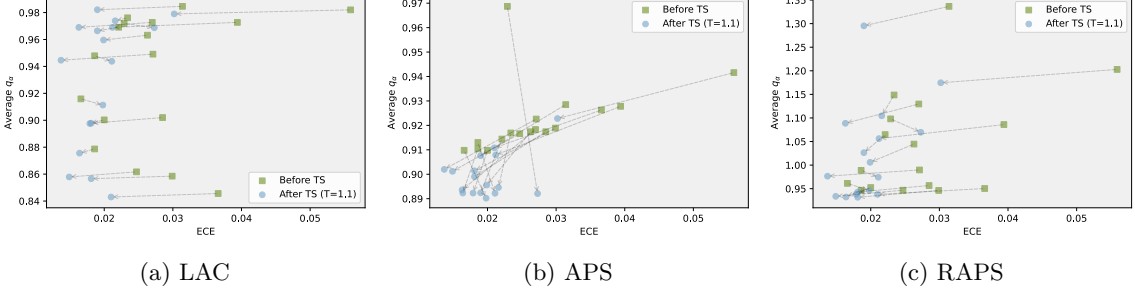

(a) LAC    (b) APS    (c) RAPS

Figure 8: **ECE and average $q_\alpha$** threshold before and after calibration for (a) LAC, (b) APS, and (c) RAPS on ImageNet. Each pair represents one model before (*square*) and after (*circle*) calibration.

Table 7: **Linear probing performance and conformal metrics for ImageNet.** Linear probing F1 score, and corresponding set size and MCCC for LAC, APS, and RAPS ($\alpha = 0.1$).

| | **F1** ($\uparrow$) | **Set size** ($\downarrow$) | | | **MCCC** ($\uparrow$) | | |
|---|---|---|---|---|---|---|---|
| | | LAC | APS | RAPS | LAC | APS | RAPS |
| DINO-S | 0.7492 | 3.27 ± 0.01 | 10.02 ± 0.02 | 4.23 ± 0.02 | 0.433 ± 0.005 | 0.477 ± 0.006 | 0.412 ± 0.005 |
| DINO-B | 0.7743 | 2.58 ± 0.00 | 10.73 ± 0.02 | 3.03 ± 0.00 | 0.416 ± 0.005 | 0.521 ± 0.005 | 0.391 ± 0.005 |
| DINOv2-S | 0.7948 | 1.87 ± 0.00 | 6.66 ± 0.01 | 2.12 ± 0.00 | 0.370 ± 0.006 | 0.520 ± 0.005 | 0.384 ± 0.006 |
| DINOv2-B | 0.8393 | 1.36 ± 0.00 | 7.50 ± 0.02 | 1.67 ± 0.00 | 0.414 ± 0.005 | 0.476 ± 0.005 | 0.489 ± 0.005 |
| DINOv2-L | 0.8624 | 1.21 ± 0.00 | 6.77 ± 0.02 | 1.50 ± 0.00 | 0.326 ± 0.004 | 0.502 ± 0.005 | 0.470 ± 0.004 |
| DINOv2-G | 0.8666 | 1.18 ± 0.00 | 4.08 ± 0.01 | 1.44 ± 0.00 | 0.244 ± 0.005 | 0.458 ± 0.006 | 0.350 ± 0.005 |
| VICReg (ResNet-50) | 0.7207 | 4.24 ± 0.01 | 14.22 ± 0.03 | 5.73 ± 0.01 | 0.472 ± 0.004 | 0.566 ± 0.004 | 0.453 ± 0.005 |
| VICReg (ResNet-50x2) | 0.7570 | 3.08 ± 0.01 | 13.49 ± 0.03 | 3.89 ± 0.01 | 0.442 ± 0.004 | 0.549 ± 0.005 | 0.445 ± 0.004 |
| VICReg (ResNet-200x2) | 0.7804 | 2.50 ± 0.00 | 12.05 ± 0.03 | 2.98 ± 0.00 | 0.440 ± 0.004 | 0.567 ± 0.004 | 0.462 ± 0.004 |
| MetaCLIP | 0.7580 | 2.39 ± 0.00 | 9.43 ± 0.02 | 2.84 ± 0.00 | 0.479 ± 0.005 | 0.535 ± 0.005 | 0.467 ± 0.004 |
| Phi 3.5 | 0.7325 | 2.93 ± 0.00 | 10.89 ± 0.02 | 3.50 ± 0.01 | 0.449 ± 0.005 | 0.535 ± 0.005 | 0.464 ± 0.005 |
| LLaVa | 0.8500 | 1.27 ± 0.00 | 3.83 ± 0.01 | 1.57 ± 0.00 | 0.424 ± 0.004 | 0.537 ± 0.005 | 0.506 ± 0.005 |
| CLIP (ViT-B) | 0.7201 | 3.03 ± 0.00 | 9.50 ± 0.02 | 3.73 ± 0.00 | 0.434 ± 0.005 | 0.556 ± 0.004 | 0.418 ± 0.006 |
| CLIP (ViT-L) | 0.8438 | 1.34 ± 0.00 | 4.40 ± 0.01 | 1.66 ± 0.00 | 0.358 ± 0.006 | 0.528 ± 0.005 | 0.462 ± 0.006 |
| CLIP (ViT-H) | 0.8393 | 1.40 ± 0.00 | 7.85 ± 0.02 | 1.75 ± 0.00 | 0.391 ± 0.007 | 0.553 ± 0.004 | 0.457 ± 0.006 |
| CLIP (ConvNeXt) | 0.7787 | 2.01 ± 0.00 | 7.92 ± 0.02 | 2.28 ± 0.00 | 0.406 ± 0.007 | 0.532 ± 0.005 | 0.454 ± 0.006 |
| CLIP (ConvNeXt-L) | 0.8165 | 1.56 ± 0.00 | 5.31 ± 0.01 | 1.88 ± 0.00 | 0.452 ± 0.006 | 0.517 ± 0.006 | 0.456 ± 0.006 |

Table 8: **Quantitative impact of calibrating vision foundation models with temperature scaling.** Average size and minimum class-conditional coverage are reported between uncalibrated/calibrated models ($T = 1$ and $T = 1.1$ respectively), with ImageNet as benchmark dataset.

| | **ΔECE** ($\times 10^{-2}$) | **Set size** ($\downarrow$) | | | **MCCC** ($\uparrow$) | | |
|---|---|---|---|---|---|---|---|
| | | LAC | APS | RAPS | LAC | APS | RAPS |
| DINO-S | 2.57 | 3.27/3.26 | 10.02/11.78 | 4.23/3.99 | 0.433/0.427 | 0.477/0.480 | 0.412/0.415 |
| DINO-B | 1.82 | 2.58/2.55 | 10.73/13.14 | 3.03/3.03 | 0.416/0.398 | 0.521/0.517 | 0.391/0.409 |
| DINOv2-S | 1.34 | 1.87/1.87 | 6.66/8.34 | 2.12/2.19 | 0.370/0.351 | 0.520/0.538 | 0.384/0.409 |
| DINOv2-B | 1.04 | 1.36/1.36 | 7.50/10.46 | 1.67/1.77 | 0.414/0.406 | 0.476/0.482 | 0.489/0.498 |
| DINOv2-L | 1.18 | 1.21/1.21 | 6.77/9.67 | 1.50/1.59 | 0.326/0.336 | 0.502/0.522 | 0.453/0.487 |
| DINOv2-G | 1.56 | 1.18/1.18 | 4.08/5.69 | 1.44/1.50 | 0.244/0.243 | 0.458/0.484 | 0.350/0.375 |
| VICReg (RN-50) | 1.22 | 4.24/4.19 | 14.22/16.60 | 5.73/5.49 | 0.472/0.468 | 0.566/0.570 | 0.453/0.456 |
| VICReg (RN-50x2) | 0.18 | 3.08/3.09 | 13.49/16.06 | 3.89/3.80 | 0.442/0.430 | 0.549/0.557 | 0.445/0.446 |
| VICReg (RN-200x2) | 0.31 | 2.50/2.49 | 12.05/14.61 | 2.98/3.01 | 0.440/0.430 | 0.567/0.566 | 0.462/0.471 |
| MetaCLIP | 0.64 | 2.39/2.40 | 9.43/11.31 | 2.84/2.84 | 0.479/0.477 | 0.535/0.541 | 0.467/0.469 |
| Phi 3.5 | -0.44 | 2.93/2.92 | 10.89/12.82 | 3.50/3.38 | 0.449/0.446 | 0.535/0.540 | 0.464/0.468 |
| LLaVa | 0.98 | 1.27/1.27 | 3.83/4.94 | 1.57/1.63 | 0.424/0.426 | 0.537/0.554 | 0.506/0.517 |
| CLIP (ViT-B) | 1.07 | 3.03/3.01 | 9.50/11.11 | 3.73/3.53 | 0.434/0.441 | 0.556/0.564 | 0.418/0.414 |
| CLIP (ViT-L) | 0.22 | 1.34/1.34 | 4.40/5.68 | 1.66/1.74 | 0.358/0.368 | 0.528/0.546 | 0.462/0.470 |
| CLIP (ViT-H) | 0.21 | 1.40/1.41 | 7.85/9.55 | 1.75/1.83 | 0.391/0.392 | 0.553/0.584 | 0.457/0.470 |
| CLIP (ConvNeXt) | -0.25 | 2.01/2.01 | 7.92/9.74 | 2.28/2.36 | 0.406/0.406 | 0.532/0.560 | 0.454/0.461 |
| CLIP (ConvNeXt-L) | -0.22 | 1.56/1.56 | 5.31/6.69 | 1.88/1.96 | 0.452/0.450 | 0.517/0.542 | 0.456/0.476 |

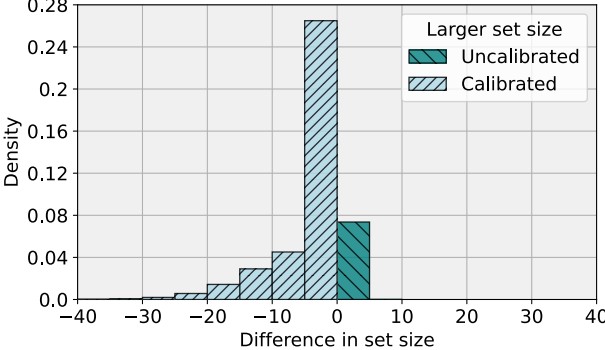

Figure 9: **Difference in conformal set size** (i.e., *efficiency*) when applying temperature scaling ($T = 1.1$) on APS and CLIP (ViT-B).

Table 9: **Quantitative impact of calibrating vision foundation models with histogram binning.** Average size and minimum class-conditional coverage are reported between uncalibrated/calibrated models, with ImageNet as benchmark dataset.

| | $\Delta$ECE | AvgSize ($\downarrow$) | | | MCCC ($\uparrow$) | | |
|---|---|---|---|---|---|---|---|
| | ($\times 10^{-2}$) | LAC | APS | RAPS | LAC | APS | RAPS |
| DINO-S | 5.59 | 3.27/135.41 | 10.02/312.12 | 4.23/312.15 | 0.433/0.422 | 0.477/0.375 | 0.412/0.375 |
| DINO-B | 3.94 | 2.58/101.94 | 10.73/269.62 | 3.03/269.67 | 0.416/0.417 | 0.521/0.388 | 0.391/0.388 |
| DINOv2-S | 2.71 | 1.87/63.19 | 6.66/186.38 | 2.12/186.44 | 0.370/0.427 | 0.520/0.404 | 0.384/0.404 |
| DINOv2-B | 2.85 | 1.36/27.33 | 7.50/87.56 | 1.67/87.68 | 0.414/0.372 | 0.476/0.414 | 0.489/0.414 |
| DINOv2-L | 2.99 | 1.21/8.86 | 6.77/31.93 | 1.50/32.06 | 0.326/0.311 | 0.502/0.391 | 0.453/0.391 |
| DINOv2-G | 3.66 | 1.18/6.25 | 4.08/20.88 | 1.44/20.98 | 0.244/0.280 | 0.458/0.386 | 0.350/0.385 |
| VICReg (RN-50) | 3.14 | 4.24/161.07 | 14.22/374.82 | 5.73/374.84 | 0.472/0.456 | 0.566/0.362 | 0.453/0.362 |
| VICReg (RN-50x2) | 2.34 | 3.08/130.26 | 13.49/322.95 | 3.89/323.00 | 0.442/0.450 | 0.549/0.397 | 0.445/0.397 |
| VICReg (RN-200x2) | 2.21 | 2.50/101.10 | 12.05/268.30 | 2.98/268.37 | 0.440/0.444 | 0.567/0.384 | 0.462/0.384 |
| MetaCLIP | 2.63 | 2.39/110.80 | 9.43/286.46 | 2.84/286.51 | 0.479/0.446 | 0.535/0.396 | 0.467/0.396 |
| Phi 3.5 | 2.29 | 2.93/140.27 | 10.89/329.86 | 3.50/329.87 | 0.449/0.456 | 0.535/0.384 | 0.464/0.385 |
| LLaVa | 2.47 | 1.27/13.72 | 3.83/47.75 | 1.57/47.87 | 0.424/0.380 | 0.537/0.401 | 0.506/0.402 |
| CLIP (ViT-B) | 2.70 | 3.03/148.29 | 9.50/337.35 | 3.73/337.35 | 0.434/0.458 | 0.556/0.374 | 0.418/0.374 |
| CLIP (ViT-L) | 1.86 | 1.34/24.29 | 4.40/74.14 | 1.66/74.21 | 0.358/0.402 | 0.528/0.418 | 0.462/0.418 |
| CLIP (ViT-H) | 2.00 | 1.40/30.29 | 7.85/99.07 | 1.75/99.19 | 0.391/0.405 | 0.553/0.427 | 0.457/0.427 |
| CLIP (ConvNeXt) | 1.86 | 2.01/84.08 | 7.92/231.47 | 2.28/231.52 | 0.406/0.449 | 0.532/0.407 | 0.454/0.407 |
| CLIP (ConvNeXt-L) | 1.66 | 1.56/42.62 | 5.31/139.04 | 1.88/139.14 | 0.452/0.435 | 0.517/0.413 | 0.456/0.413 |

Table 10: **Evaluation under domain-shift.** Set size ($\downarrow$), coverage ($\uparrow$), and MCCC ($\uparrow$) across three CP methods and three foundation models.

| | | Set size ($\downarrow$) | | | Coverage ($\uparrow$) | | | MCCC ($\uparrow$) | | |
|---|---|---|---|---|---|---|---|---|---|---|
| | | LAC | APS | RAPS | LAC | APS | RAPS | LAC | APS | RAPS |
| ImageNet | DINOv2-B | 1.36 ± 0.00 | 7.50 ± 0.02 | 1.67 ± 0.00 | 0.900 ± 0.000 | 0.900 ± 0.000 | 0.900 ± 0.000 | 0.414 ± 0.005 | 0.476 ± 0.005 | 0.489 ± 0.005 |
| | VICReg (RN 50x2) | 3.08 ± 0.01 | 13.49 ± 0.02 | 3.89 ± 0.01 | 0.900 ± 0.000 | 0.900 ± 0.000 | 0.900 ± 0.000 | 0.442 ± 0.004 | 0.549 ± 0.005 | 0.445 ± 0.005 |
| | CLIP (ViT-B) | 3.03 ± 0.00 | 9.50 ± 0.02 | 3.73 ± 0.00 | 0.900 ± 0.000 | 0.901 ± 0.000 | 0.900 ± 0.000 | 0.434 ± 0.005 | 0.556 ± 0.004 | 0.418 ± 0.006 |
| ImageNet-V2 | DINOv2-B | 1.38 ± 0.00 | 9.47 ± 0.04 | 1.74 ± 0.00 | 0.879 ± 0.000 | 0.892 ± 0.000 | 0.881 ± 0.000 | 0.081 ± 0.009 | 0.187 ± 0.009 | 0.068 ± 0.009 |
| | VICReg (RN 50x2) | 3.15 ± 0.00 | 12.54 ± 0.04 | 3.89 ± 0.00 | 0.867 ± 0.000 | 0.871 ± 0.000 | 0.870 ± 0.000 | 0.130 ± 0.008 | 0.175 ± 0.009 | 0.143 ± 0.009 |
| | CLIP (ViT-B) | 2.88 ± 0.00 | 6.79 ± 0.02 | 3.58 ± 0.00 | 0.877 ± 0.000 | 0.864 ± 0.000 | 0.882 ± 0.000 | 0.023 ± 0.005 | 0.034 ± 0.007 | 0.029 ± 0.005 |
| ImageNet-Sketch | DINOv2-B | 1.35 ± 0.09 | 10.90 ± 0.11 | 1.91 ± 0.31 | 0.884 ± 0.001 | 0.904 ± 0.001 | 0.893 ± 0.001 | 0.221 ± 0.008 | 0.322 ± 0.007 | 0.275 ± 0.007 |
| | VICReg (RN 50x2) | 3.49 ± 0.01 | 21.98 ± 0.05 | 4.51 ± 0.01 | 0.826 ± 0.000 | 0.879 ± 0.000 | 0.830 ± 0.000 | 0.119 ± 0.007 | 0.223 ± 0.008 | 0.114 ± 0.007 |
| | CLIP (ViT-B) | 3.16 ± 0.01 | 12.98 ± 0.03 | 3.99 ± 0.01 | 0.877 ± 0.000 | 0.900 ± 0.000 | 0.878 ± 0.000 | 0.182 ± 0.008 | 0.293 ± 0.008 | 0.185 ± 0.007 |
| ImageNet-A | DINOv2-B | 1.63 ± 1.76 | 25.30 ± 1.58 | 3.55 ± 1.80 | 0.646 ± 0.001 | 0.897 ± 0.001 | 0.734 ± 0.001 | 0.000 ± 0.010 | 0.000 ± 0.000 | 0.000 ± 0.010 |
| | VICReg (RN 50x2) | 8.43 ± 0.03 | 59.83 ± 0.07 | 7.82 ± 0.01 | 0.531 ± 0.001 | 0.879 ± 0.001 | 0.511 ± 0.001 | 0.000 ± 0.000 | 0.000 ± 0.000 | 0.000 ± 0.000 |
| | CLIP (ViT-B) | 6.79 ± 0.02 | 39.42 ± 0.07 | 6.45 ± 0.01 | 0.648 ± 0.001 | 0.912 ± 0.001 | 0.624 ± 0.001 | 0.000 ± 0.000 | 0.000 ± 0.000 | 0.000 ± 0.000 |
| ImageNet-R | DINOv2-B | 1.31 ± 0.05 | 7.49 ± 0.04 | 1.94 ± 0.24 | 0.867 ± 0.001 | 0.910 ± 0.001 | 0.879 ± 0.001 | 0.282 ± 0.006 | 0.532 ± 0.009 | 0.408 ± 0.007 |
| | VICReg (RN 50x2) | 5.84 ± 0.02 | 27.36 ± 0.04 | 5.65 ± 0.01 | 0.761 ± 0.001 | 0.897 ± 0.000 | 0.742 ± 0.001 | 0.198 ± 0.007 | 0.429 ± 0.008 | 0.186 ± 0.007 |
| | CLIP (ViT-B) | 3.38 ± 0.01 | 11.93 ± 0.03 | 4.13 ± 0.00 | 0.893 ± 0.000 | 0.930 ± 0.000 | 0.890 ± 0.000 | 0.422 ± 0.008 | 0.604 ± 0.007 | 0.386 ± 0.009 |

Table 11: **Set size and CovGap** for CLIP (ViT-B) on ImageNet and its variants.

| | | Set size ($\downarrow$) | | | CovGap ($\downarrow$) | | |
|---|---|---|---|---|---|---|---|
| | | LAC | APS | RAPS | LAC | APS | RAPS |
| ImageNet | ZS | 2.81 ± 0.00 | 10.07 ± 0.02 | 3.23 ± 0.00 | 0.086 ± 0.000 | 0.069 ± 0.000 | 0.084 ± 0.000 |
| | ZSLP | 2.17 ± 0.00 | 6.60 ± 0.01 | 2.45 ± 0.00 | 0.076 ± 0.000 | 0.060 ± 0.000 | 0.074 ± 0.000 |
| | CoOp (Zhou et al., 2022b) | 2.44 ± 0.00 | 6.32 ± 0.01 | 2.79 ± 0.00 | 0.080 ± 0.000 | 0.066 ± 0.000 | 0.080 ± 0.000 |
| | KgCoOp (Hantao Yao, 2023) | 2.47 ± 0.00 | 7.18 ± 0.01 | 2.84 ± 0.00 | 0.081 ± 0.000 | 0.065 ± 0.000 | 0.081 ± 0.000 |
| | CLAP (Silva-Rodríguez et al., 2024) | 2.15 ± 0.00 | 6.83 ± 0.01 | 2.45 ± 0.00 | 0.077 ± 0.000 | 0.060 ± 0.000 | 0.074 ± 0.000 |
| -V2 | ZS | 4.64 ± 0.02 | 17.34 ± 0.08 | 5.72 ± 0.03 | 0.128 ± 0.000 | 0.124 ± 0.000 | 0.128 ± 0.000 |
| | ZSLP | 3.89 ± 0.01 | 12.51 ± 0.05 | 4.81 ± 0.01 | 0.126 ± 0.000 | 0.121 ± 0.000 | 0.125 ± 0.000 |
| | CoOp (Zhou et al., 2022b) | 4.08 ± 0.01 | 11.26 ± 0.06 | 4.87 ± 0.01 | 0.126 ± 0.000 | 0.123 ± 0.000 | 0.126 ± 0.000 |
| | KgCoOp (Hantao Yao, 2023) | 4.09 ± 0.01 | 12.65 ± 0.06 | 4.93 ± 0.02 | 0.126 ± 0.000 | 0.123 ± 0.000 | 0.126 ± 0.000 |
| | CLAP (Silva-Rodríguez et al., 2024) | 3.72 ± 0.01 | 12.72 ± 0.05 | 4.62 ± 0.01 | 0.126 ± 0.000 | 0.122 ± 0.000 | 0.125 ± 0.000 |
| -Sketch | ZS | 14.74 ± 0.02 | 37.18 ± 0.06 | 20.39 ± 0.04 | 0.099 ± 0.000 | 0.090 ± 0.000 | 0.099 ± 0.000 |
| | ZSLP | 16.46 ± 0.03 | 37.27 ± 0.05 | 22.79 ± 0.05 | 0.098 ± 0.000 | 0.091 ± 0.000 | 0.098 ± 0.000 |
| | CoOp (Zhou et al., 2022b) | 16.06 ± 0.03 | 34.36 ± 0.05 | 22.12 ± 0.05 | 0.098 ± 0.000 | 0.091 ± 0.000 | 0.099 ± 0.000 |
| | KgCoOp (Hantao Yao, 2023) | 15.46 ± 0.02 | 34.35 ± 0.05 | 21.74 ± 0.04 | 0.098 ± 0.000 | 0.092 ± 0.000 | 0.099 ± 0.000 |
| | CLAP (Silva-Rodríguez et al., 2024) | 15.20 ± 0.03 | 34.90 ± 0.05 | 21.29 ± 0.05 | 0.098 ± 0.000 | 0.090 ± 0.000 | 0.098 ± 0.000 |
| -A | ZS | 9.42 ± 0.04 | 16.59 ± 0.06 | 11.28 ± 0.05 | 0.094 ± 0.000 | 0.085 ± 0.000 | 0.095 ± 0.000 |
| | ZSLP | 10.92 ± 0.04 | 18.63 ± 0.07 | 12.47 ± 0.05 | 0.087 ± 0.000 | 0.080 ± 0.000 | 0.088 ± 0.000 |
| | CoOp (Zhou et al., 2022b) | 9.23 ± 0.03 | 14.65 ± 0.05 | 10.72 ± 0.05 | 0.087 ± 0.001 | 0.084 ± 0.000 | 0.090 ± 0.000 |
| | KgCoOp (Hantao Yao, 2023) | 8.66 ± 0.04 | 14.39 ± 0.06 | 10.19 ± 0.04 | 0.093 ± 0.000 | 0.084 ± 0.000 | 0.096 ± 0.000 |
| | CLAP (Silva-Rodríguez et al., 2024) | 9.36 ± 0.03 | 16.56 ± 0.06 | 11.38 ± 0.04 | 0.087 ± 0.000 | 0.082 ± 0.000 | 0.087 ± 0.000 |
| -R | ZS | 1.92 ± 0.00 | 5.77 ± 0.01 | 2.31 ± 0.00 | 0.059 ± 0.000 | 0.039 ± 0.000 | 0.054 ± 0.000 |
| | ZSLP | 2.07 ± 0.00 | 5.62 ± 0.01 | 2.36 ± 0.00 | 0.058 ± 0.000 | 0.041 ± 0.000 | 0.056 ± 0.000 |
| | CoOp (Zhou et al., 2022b) | 2.06 ± 0.00 | 5.11 ± 0.01 | 2.40 ± 0.00 | 0.061 ± 0.000 | 0.042 ± 0.000 | 0.063 ± 0.000 |
| | KgCoOp (Hantao Yao, 2023) | 1.96 ± 0.00 | 5.32 ± 0.01 | 2.29 ± 0.00 | 0.060 ± 0.000 | 0.041 ± 0.000 | 0.058 ± 0.000 |
| | CLAP (Silva-Rodríguez et al., 2024) | 1.92 ± 0.00 | 5.46 ± 0.01 | 2.26 ± 0.01 | 0.059 ± 0.000 | 0.039 ± 0.000 | 0.053 ± 0.000 |

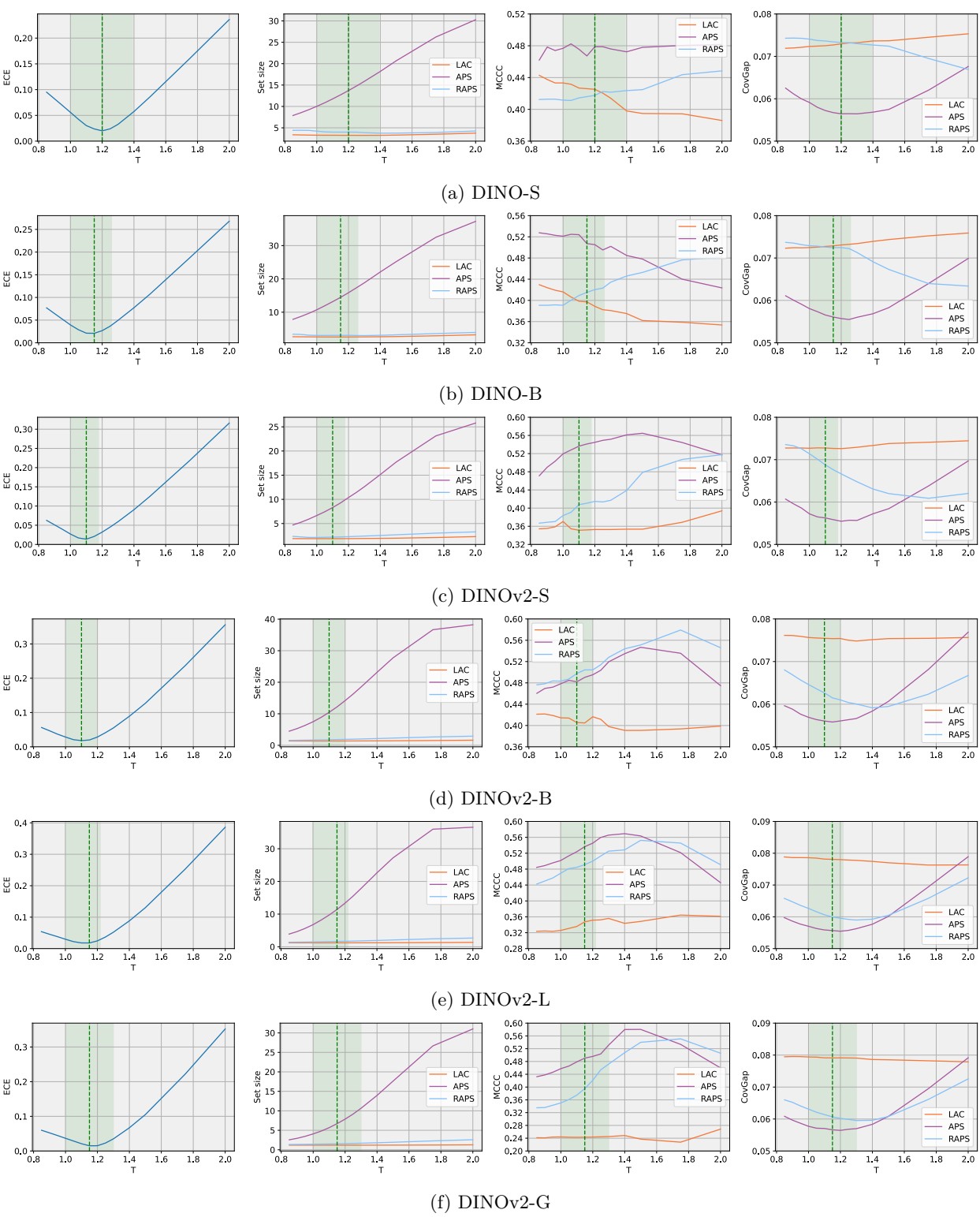

Figure 10: **Impact of the temperature $T$** on the ECE, set size, MCCC, and CovGap, for the DINO and DINOv2 models on ImageNet. $T = 1$ indicates the uncalibrated model performance. Green vertical line indicates the model performance for the optimal temperature.

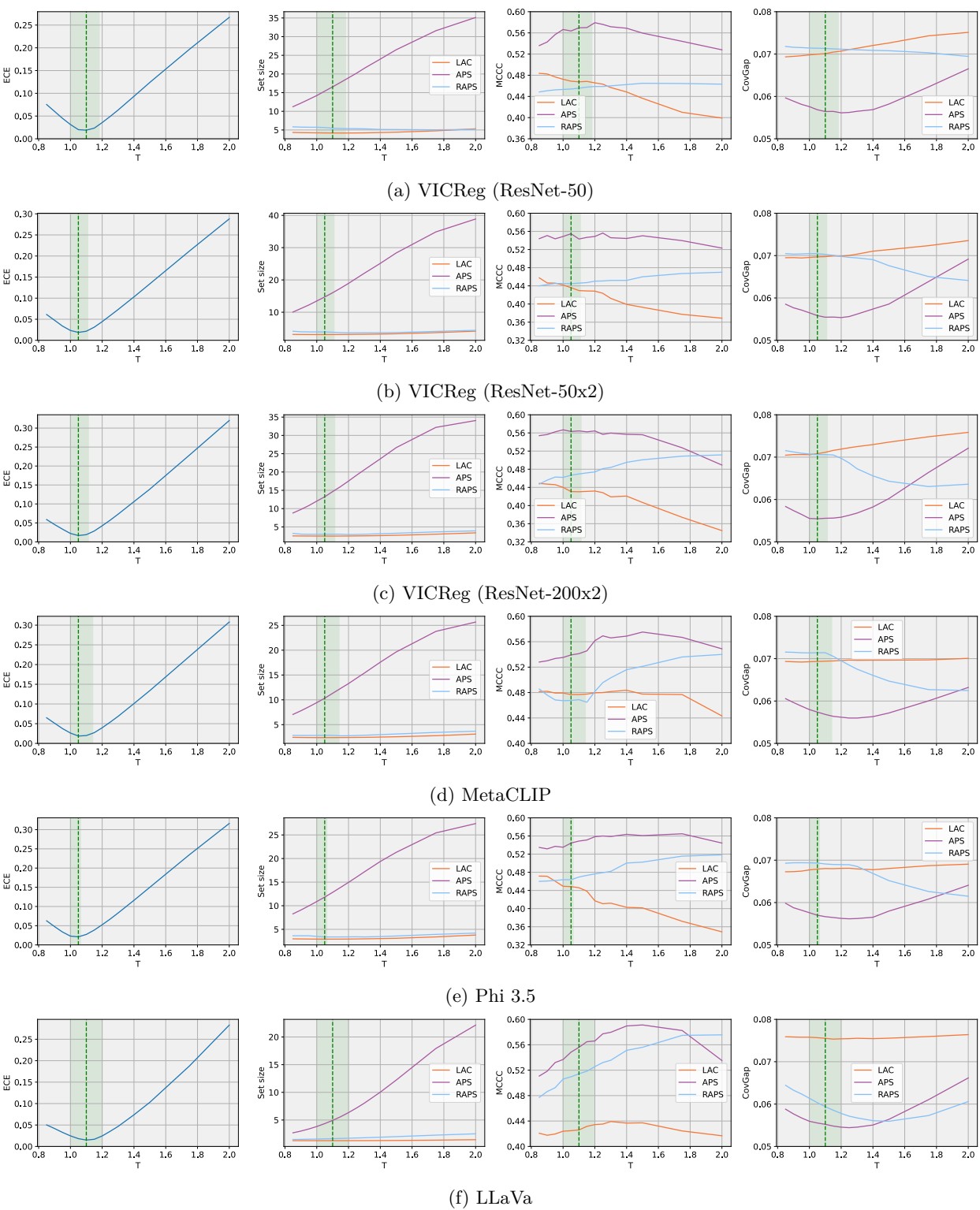

Figure 11: **Impact of the temperature $T$** on the ECE, set size, MCCC, and CovGap, for the VICReg, MetaCLIP, Phi 3.5 and LLaVa models on ImageNet. $T = 1$ indicates the uncalibrated model performance. Green vertical line indicates the model performance for the optimal temperature.

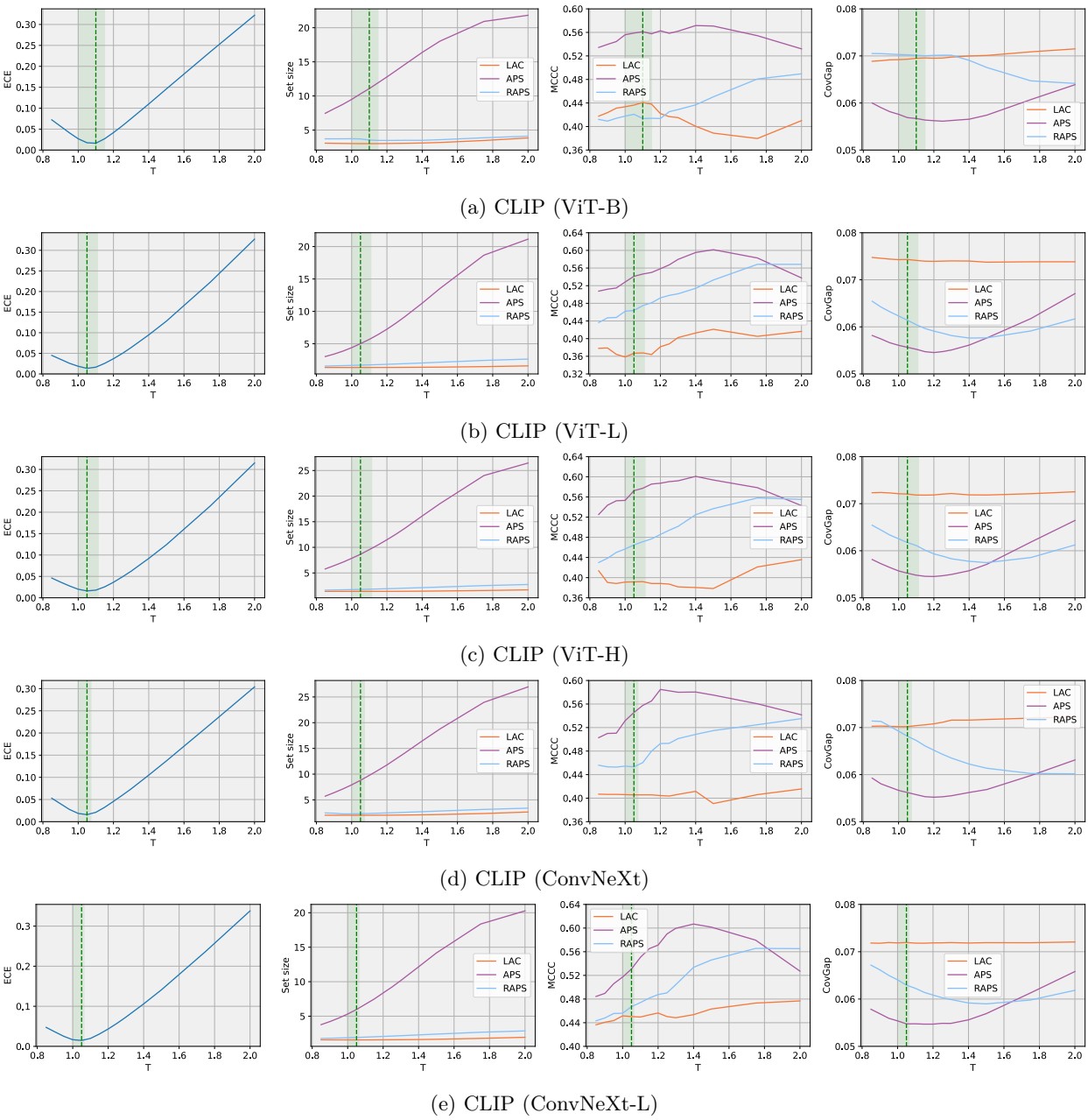

Figure 12: **Impact of the temperature $T$** on the ECE, set size, MCCC, and CovGap, for the CLIP models on ImageNet. $T = 1$ indicates the uncalibrated model performance. Green vertical line indicates the model performance for the optimal temperature.

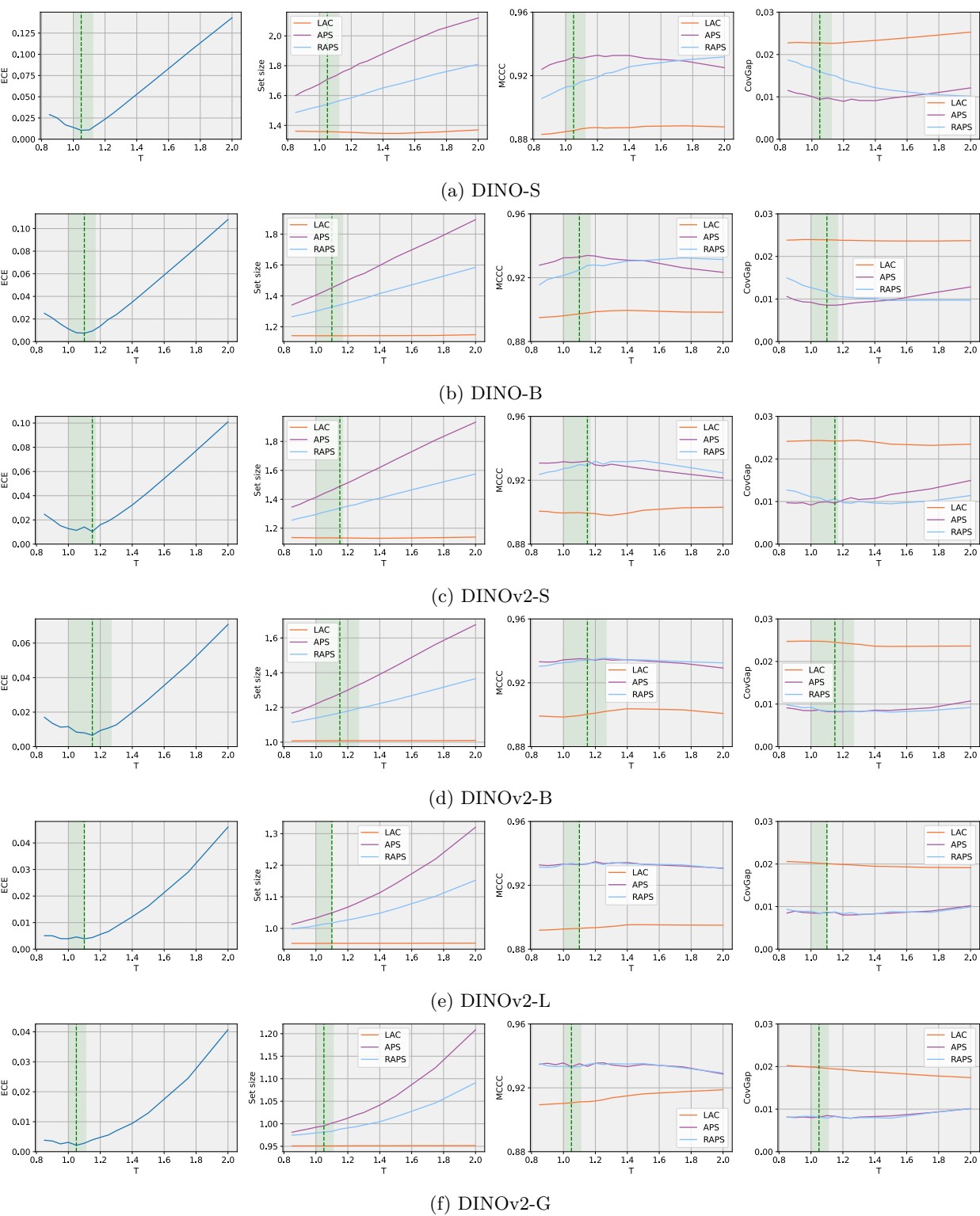

Figure 13: **Impact of the temperature $T$** on the ECE, set size, MCCC, and CovGap, for the DINO and DINOv2 models on CIFAR-10. $T = 1$ indicates the uncalibrated model performance. Green vertical line indicates the model performance for the optimal temperature.

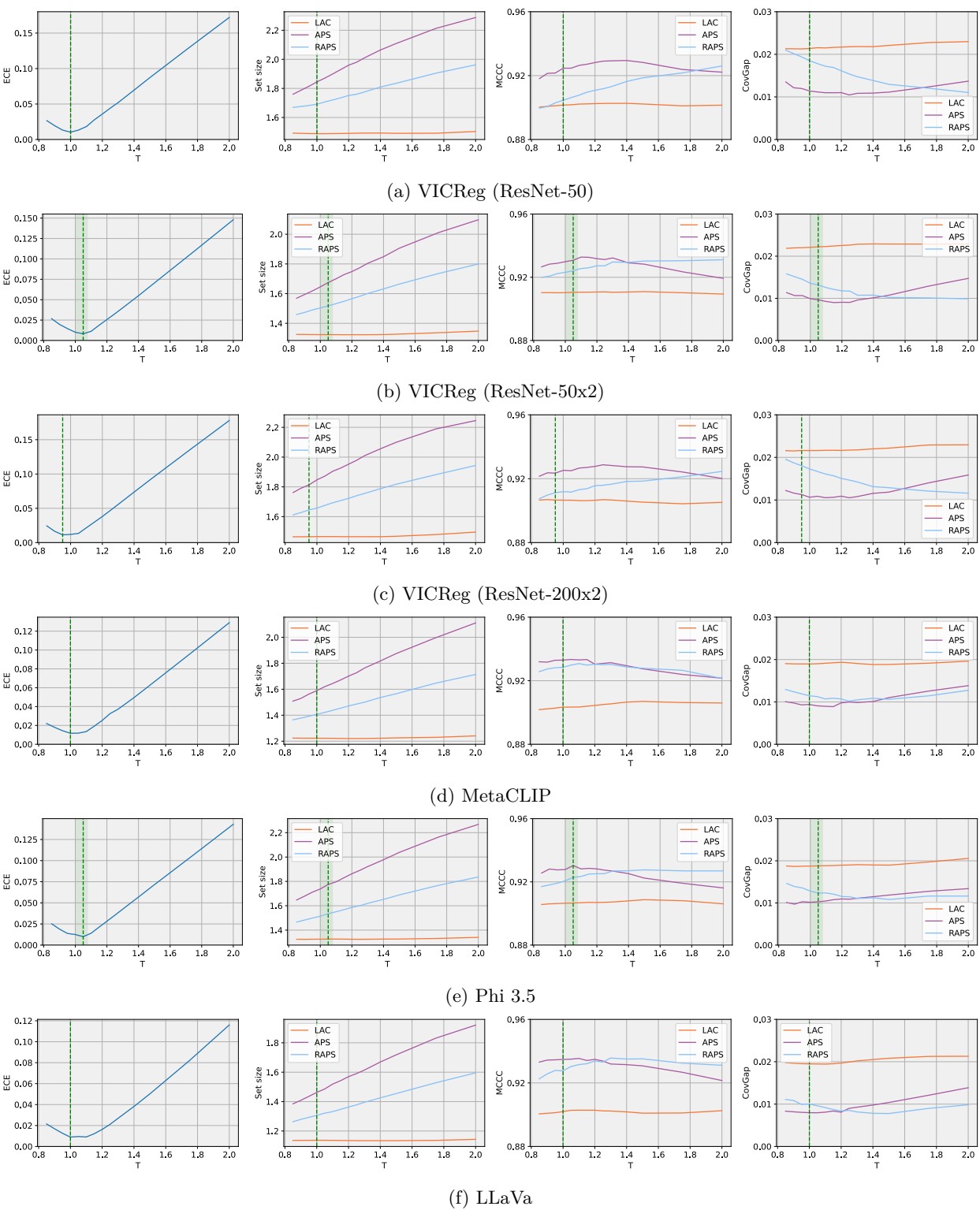

Figure 14: **Impact of the temperature $T$** on the ECE, set size, MCCC, and CovGap, for the VICReg, MetaCLIP, Phi 3.5 and LLaVa models on CIFAR-10. $T = 1$ indicates the uncalibrated model performance. Green vertical line indicates the model performance for the optimal temperature.

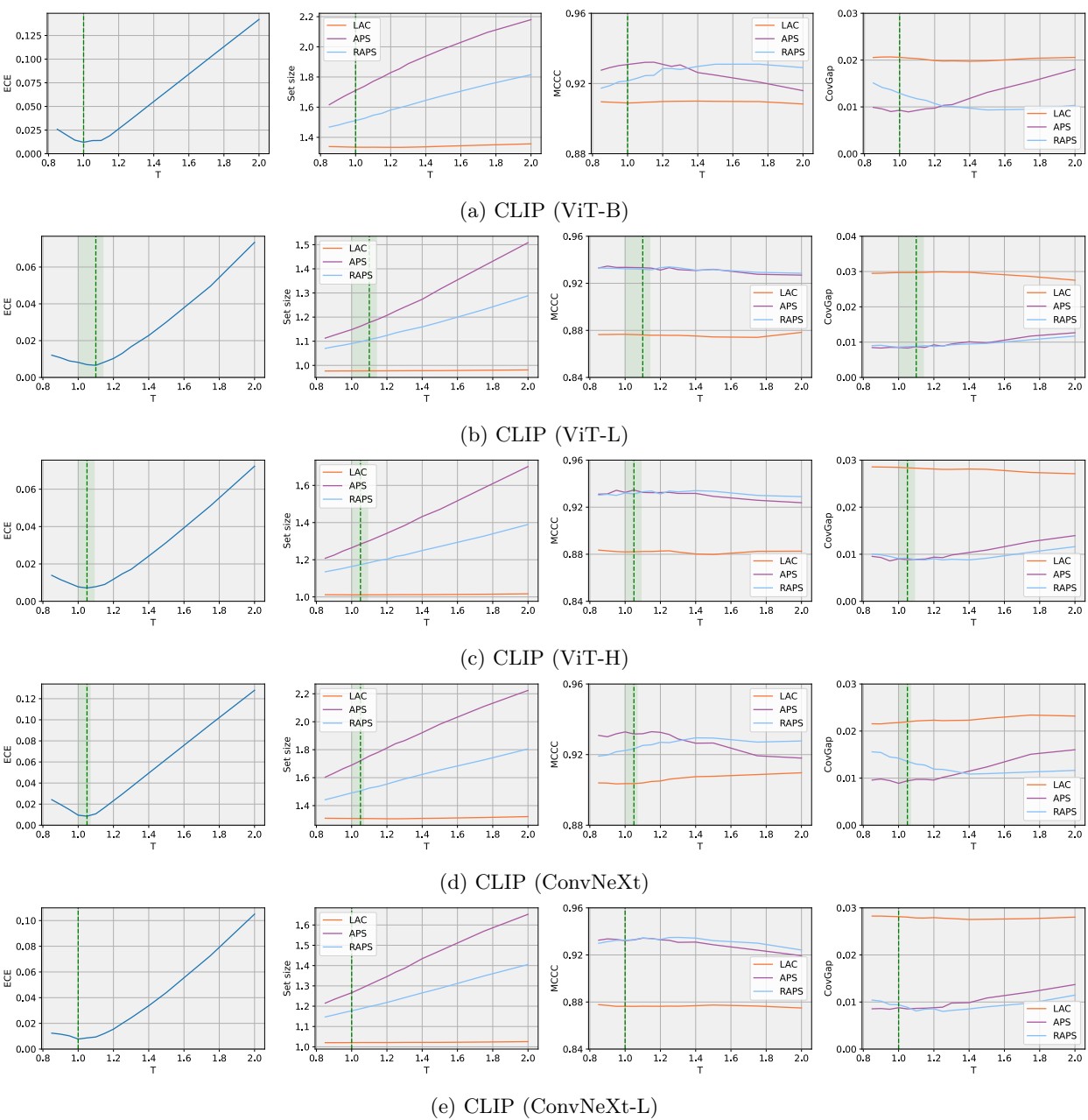

(a) CLIP (ViT-B)

(b) CLIP (ViT-L)

(c) CLIP (ViT-H)

(d) CLIP (ConvNeXt)

(e) CLIP (ConvNeXt-L)

Figure 15: **Impact of the temperature $T$** on the ECE, set size, MCCC, and CovGap, for the CLIP models on CIFAR-10. $T = 1$ indicates the uncalibrated model performance. Green vertical line indicates the model performance for the optimal temperature.

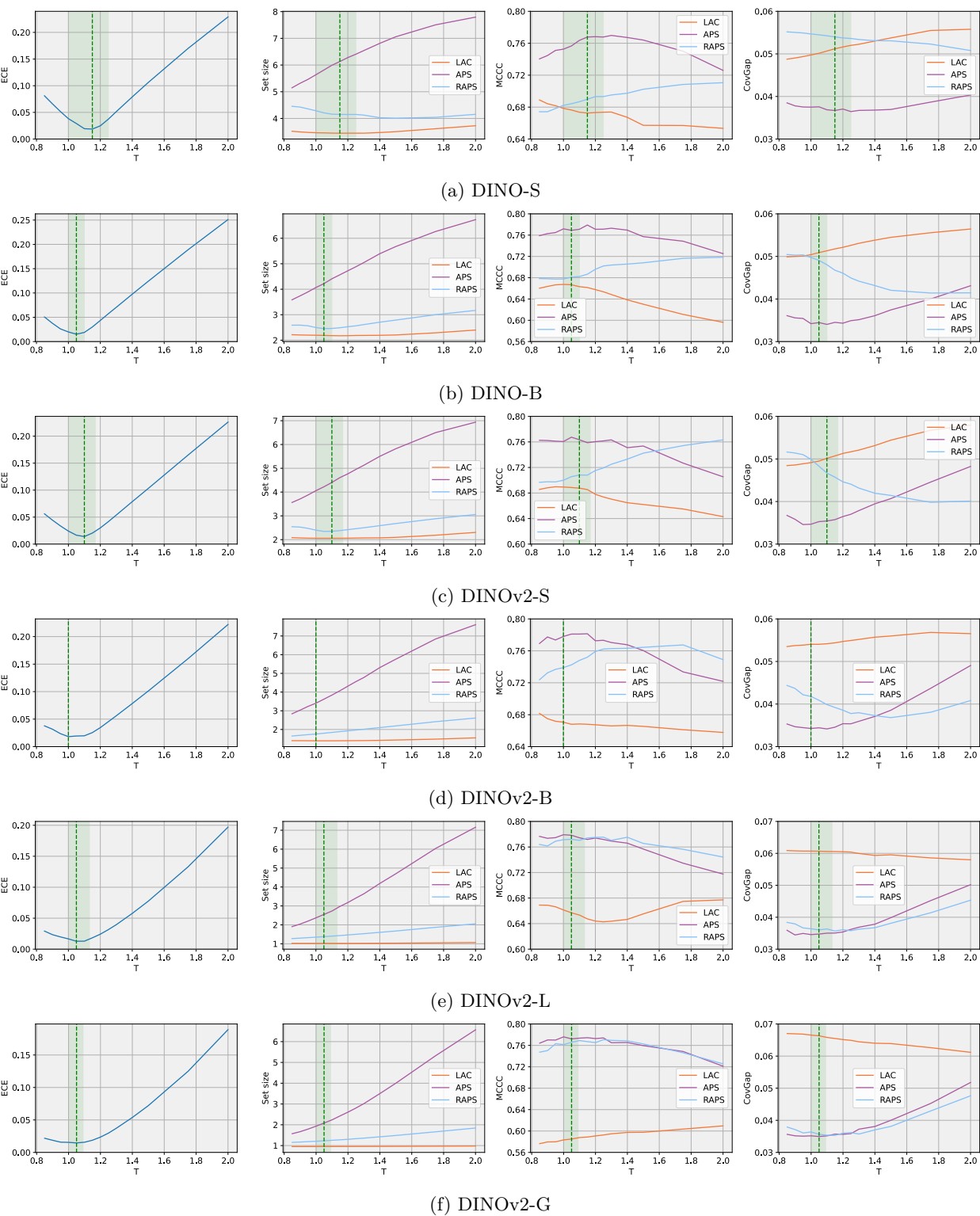

Figure 16: **Impact of the temperature $T$** on the ECE, set size, MCCC, and CovGap, for the DINO and DINOv2 models on CIFAR-100. $T = 1$ indicates the uncalibrated model performance. Green vertical line indicates the model performance for the optimal temperature.

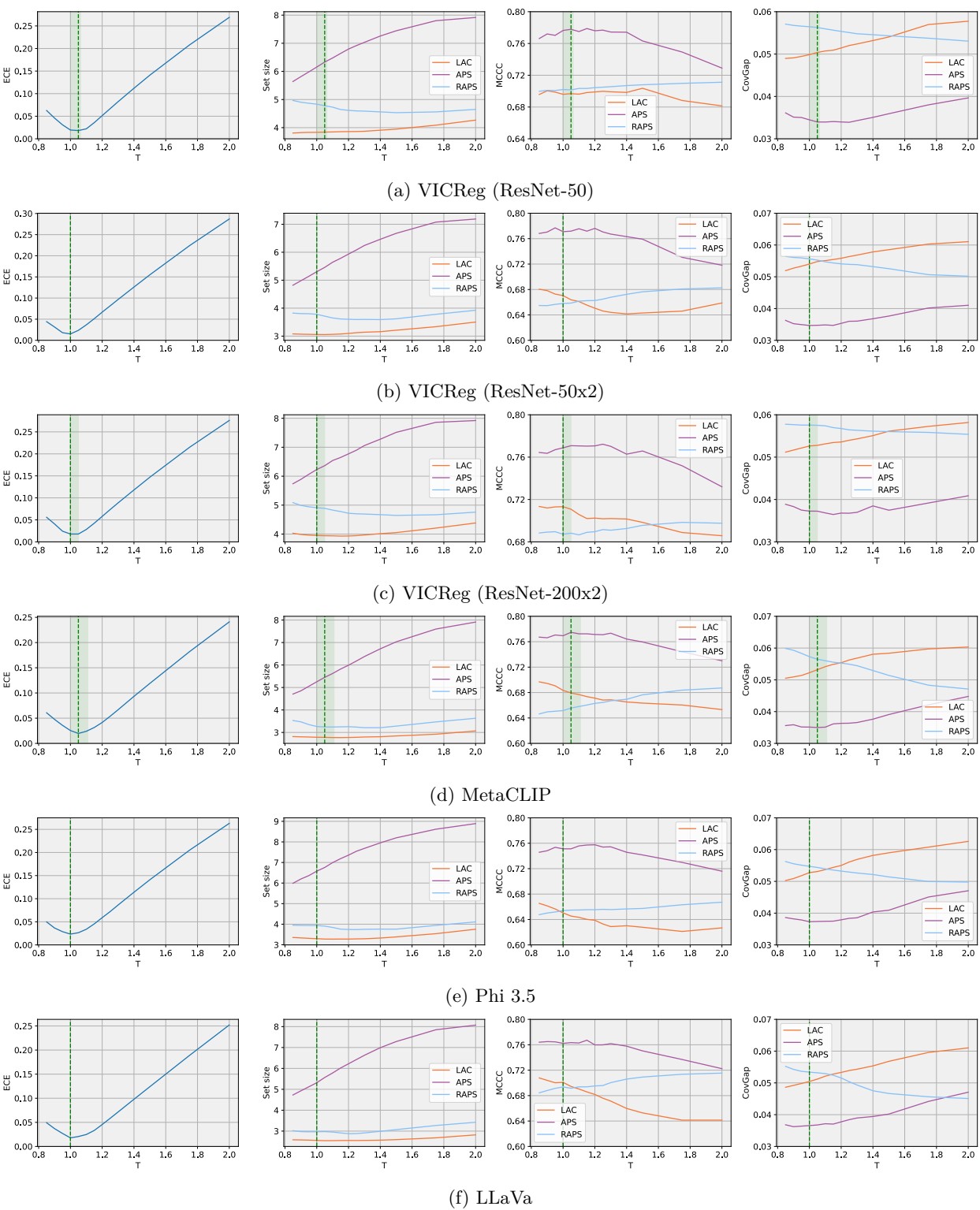

Figure 17: **Impact of the temperature $T$** on the ECE, set size, MCCC, and CovGap, for the VICReg, MetaCLIP, Phi 3.5 and LLaVa models on CIFAR-100. $T = 1$ indicates the uncalibrated model performance. Green vertical line indicates the model performance for the optimal temperature.

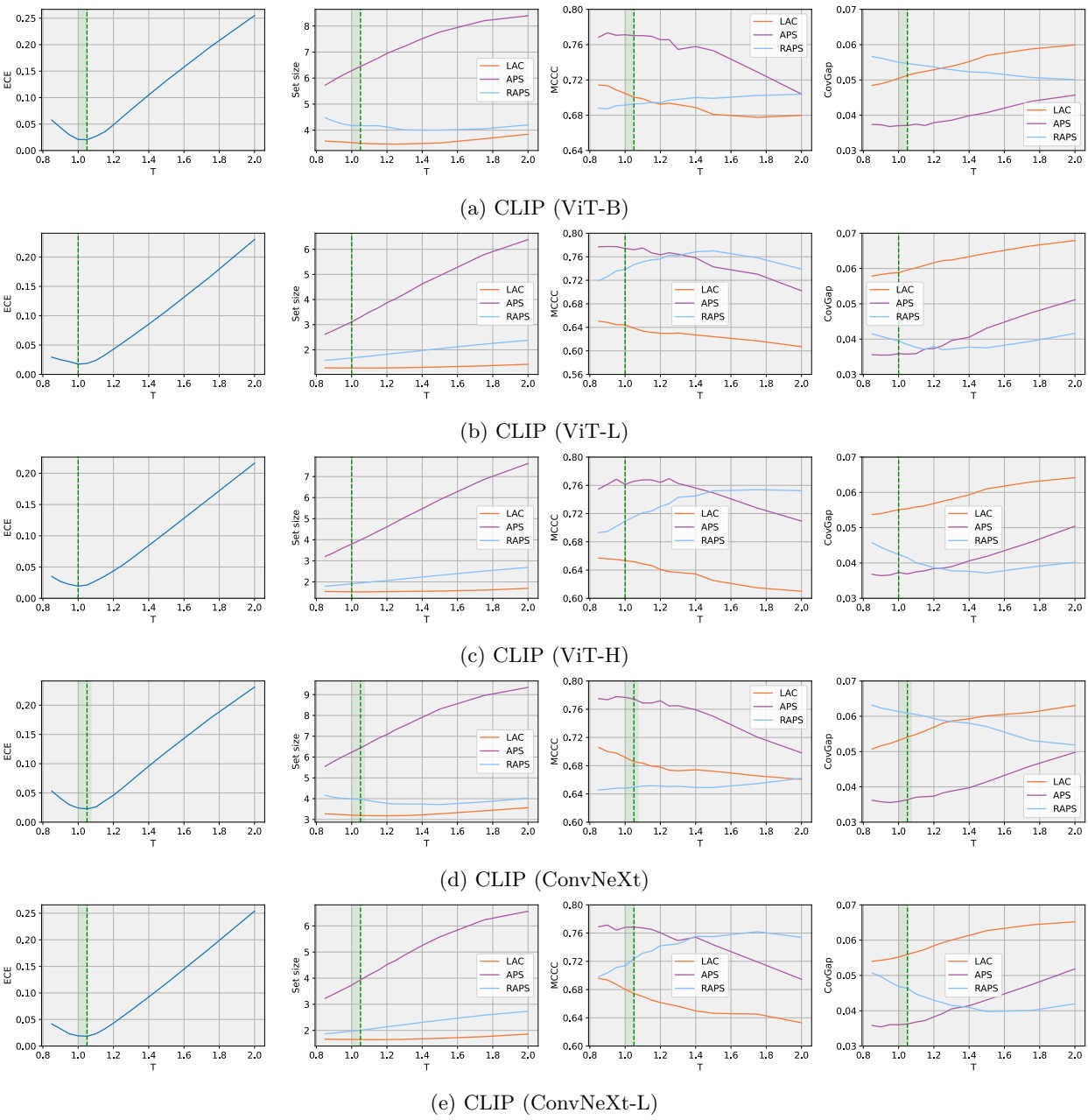

Figure 18: **Impact of the temperature $T$** on the ECE, set size, MCCC, and CovGap, for the CLIP models on CIFAR-100. $T = 1$ indicates the uncalibrated model performance. Green vertical line indicates the model performance for the optimal temperature.

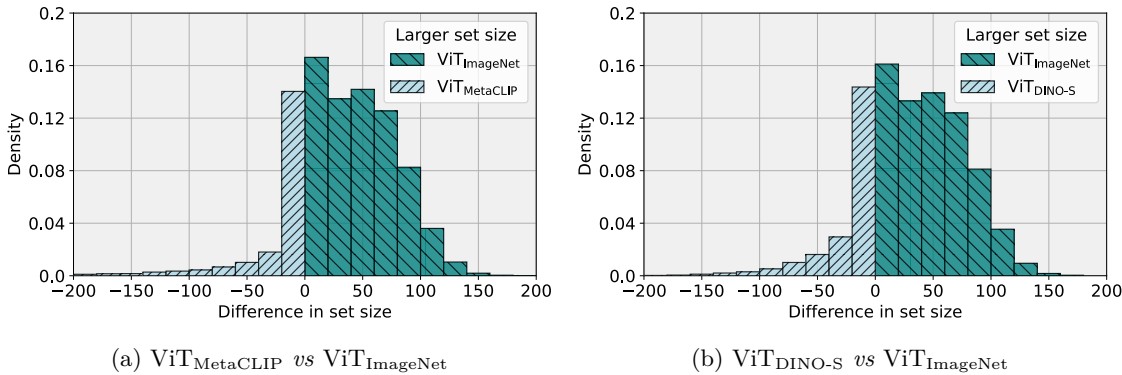

(a) ViT$_{\text{MetaCLIP}}$ *vs* ViT$_{\text{ImageNet}}$

(b) ViT$_{\text{DINO-S}}$ *vs* ViT$_{\text{ImageNet}}$

Figure 19: **Difference in set size for APS** between (a) ViT$_{\text{MetaCLIP}}$ and ViT$_{\text{ImageNet}}$ and between (b) ViT$_{\text{DINO-S}}$ and ViT$_{\text{ImageNet}}$. Equal set sizes not shown.

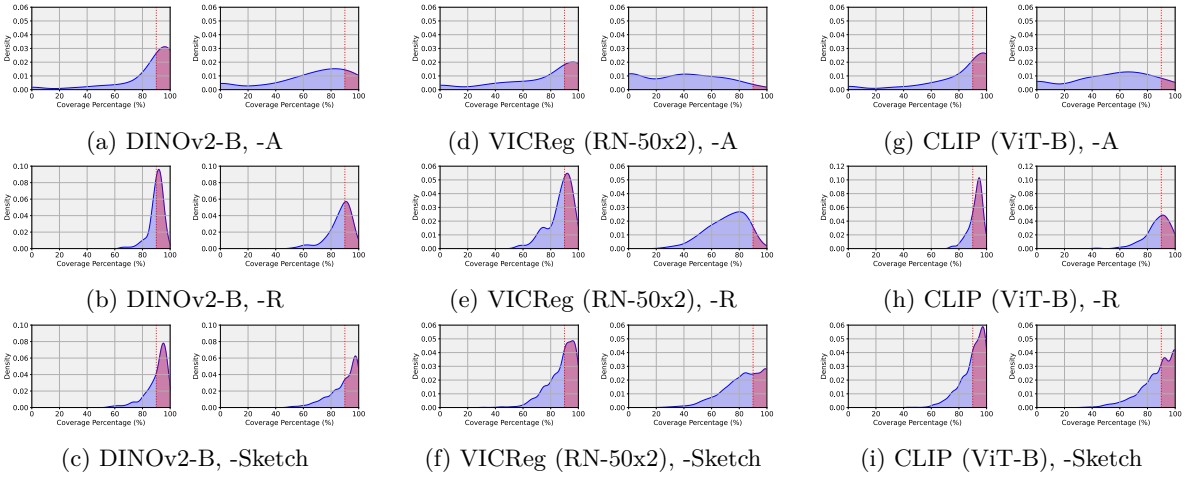

Figure 20: **Domain shift analysis.** Distribution of class-conditional coverages for (a, b, c) DINOv2-B, (d, e, f) VICReg (RN 50x2), and (g, h, i) CLIP (ViT-B). The distribution shift datasets are (a, d, g) ImageNet-A, (b, e, h) ImageNet-R, and (c, f, i) and ImageNet-Sketch. APS (*left*) and RAPS (*right*).

Table 12: **Set size and CovGap** for CLIP (ViT-B) on 11 datasets across several few-shot adaptation approaches. Training for 4 shots.

| | | Set size (↓) | | | CovGap (↓) | | |
|---|---|---|---|---|---|---|---|
| | | LAC | APS | RAPS | LAC | APS | RAPS |
| Aircraft | ZS | 18.70 ± 0.06 | 19.75 ± 0.06 | 20.74 ± 0.05 | 0.138 ± 0.000 | 0.139 ± 0.000 | 0.127 ± 0.000 |
| | ZSLP | 9.73 ± 0.03 | 11.13 ± 0.03 | 11.55 ± 0.05 | 0.082 ± 0.001 | 0.073 ± 0.000 | 0.086 ± 0.001 |
| | CoOp (Zhou et al., 2022b) | 9.41 ± 0.03 | 10.62 ± 0.04 | 11.38 ± 0.04 | 0.081 ± 0.001 | 0.076 ± 0.000 | 0.088 ± 0.001 |
| | KgCoOp (Hantao Yao, 2023) | 9.46 ± 0.03 | 10.82 ± 0.03 | 11.00 ± 0.04 | 0.084 ± 0.001 | 0.076 ± 0.000 | 0.086 ± 0.001 |
| | CLAP (Silva-Rodríguez et al., 2024) | 9.73 ± 0.03 | 11.00 ± 0.03 | 11.51 ± 0.05 | 0.082 ± 0.001 | 0.073 ± 0.000 | 0.086 ± 0.001 |
| Caltech-101 | ZS | 0.94 ± 0.00 | 1.95 ± 0.01 | 1.20 ± 0.00 | 0.136 ± 0.001 | 0.092 ± 0.001 | 0.092 ± 0.001 |
| | ZSLP | 0.91 ± 0.00 | 1.20 ± 0.00 | 1.05 ± 0.00 | 0.147 ± 0.001 | 0.085 ± 0.001 | 0.085 ± 0.001 |
| | CoOp (Zhou et al., 2022b) | 0.92 ± 0.00 | 1.14 ± 0.00 | 1.06 ± 0.00 | 0.143 ± 0.001 | 0.085 ± 0.001 | 0.087 ± 0.001 |
| | KgCoOp (Hantao Yao, 2023) | 0.92 ± 0.00 | 1.25 ± 0.00 | 1.08 ± 0.00 | 0.147 ± 0.001 | 0.084 ± 0.001 | 0.085 ± 0.001 |
| | CLAP (Silva-Rodríguez et al., 2024) | 0.91 ± 0.00 | 1.31 ± 0.01 | 1.08 ± 0.00 | 0.144 ± 0.001 | 0.085 ± 0.001 | 0.086 ± 0.001 |
| DTD | ZS | 11.78 ± 0.06 | 13.30 ± 0.04 | 13.66 ± 0.06 | 0.122 ± 0.001 | 0.117 ± 0.001 | 0.121 ± 0.001 |
| | ZSLP | 4.74 ± 0.03 | 6.83 ± 0.03 | 5.67 ± 0.04 | 0.092 ± 0.001 | 0.077 ± 0.001 | 0.091 ± 0.001 |
| | CoOp (Zhou et al., 2022b) | 5.68 ± 0.03 | 6.46 ± 0.04 | 7.00 ± 0.05 | 0.099 ± 0.001 | 0.093 ± 0.001 | 0.098 ± 0.001 |
| | KgCoOp (Hantao Yao, 2023) | 5.33 ± 0.03 | 6.22 ± 0.04 | 6.33 ± 0.05 | 0.097 ± 0.001 | 0.083 ± 0.001 | 0.096 ± 0.001 |
| | CLAP (Silva-Rodríguez et al., 2024) | 4.75 ± 0.03 | 6.83 ± 0.03 | 5.68 ± 0.04 | 0.094 ± 0.001 | 0.079 ± 0.001 | 0.093 ± 0.001 |
| EuroSAT | ZS | 4.55 ± 0.01 | 4.78 ± 0.00 | 5.06 ± 0.01 | 0.103 ± 0.001 | 0.101 ± 0.001 | 0.102 ± 0.000 |
| | ZSLP | 1.62 ± 0.00 | 2.22 ± 0.00 | 1.99 ± 0.00 | 0.043 ± 0.000 | 0.051 ± 0.000 | 0.042 ± 0.000 |
| | CoOp (Zhou et al., 2022b) | 1.82 ± 0.00 | 2.16 ± 0.00 | 1.95 ± 0.00 | 0.043 ± 0.000 | 0.028 ± 0.000 | 0.034 ± 0.000 |
| | KgCoOp (Hantao Yao, 2023) | 1.51 ± 0.00 | 1.92 ± 0.00 | 1.76 ± 0.00 | 0.028 ± 0.000 | 0.030 ± 0.000 | 0.032 ± 0.000 |
| | CLAP (Silva-Rodríguez et al., 2024) | 1.73 ± 0.00 | 2.33 ± 0.00 | 2.06 ± 0.00 | 0.043 ± 0.000 | 0.048 ± 0.000 | 0.041 ± 0.000 |
| Flowers-102 | ZS | 9.26 ± 0.05 | 9.52 ± 0.06 | 15.15 ± 0.08 | 0.180 ± 0.000 | 0.175 ± 0.000 | 0.179 ± 0.000 |
| | ZSLP | 1.09 ± 0.00 | 2.36 ± 0.01 | 1.65 ± 0.00 | 0.132 ± 0.001 | 0.089 ± 0.001 | 0.095 ± 0.001 |
| | CoOp (Zhou et al., 2022b) | 1.13 ± 0.00 | 1.80 ± 0.01 | 1.51 ± 0.00 | 0.104 ± 0.001 | 0.080 ± 0.001 | 0.083 ± 0.001 |
| | KgCoOp (Hantao Yao, 2023) | 1.19 ± 0.00 | 2.04 ± 0.01 | 1.64 ± 0.00 | 0.114 ± 0.001 | 0.082 ± 0.001 | 0.085 ± 0.001 |
| | CLAP (Silva-Rodríguez et al., 2024) | 1.15 ± 0.00 | 2.47 ± 0.01 | 1.70 ± 0.00 | 0.137 ± 0.001 | 0.092 ± 0.001 | 0.097 ± 0.001 |
| Food-101 | ZS | 1.14 ± 0.00 | 1.89 ± 0.00 | 1.45 ± 0.00 | 0.054 ± 0.000 | 0.026 ± 0.000 | 0.031 ± 0.000 |
| | ZSLP | 1.11 ± 0.00 | 1.82 ± 0.00 | 1.42 ± 0.00 | 0.046 ± 0.000 | 0.023 ± 0.000 | 0.026 ± 0.000 |
| | CoOp (Zhou et al., 2022b) | 1.15 ± 0.00 | 1.77 ± 0.00 | 1.41 ± 0.00 | 0.050 ± 0.000 | 0.026 ± 0.000 | 0.030 ± 0.000 |
| | KgCoOp (Hantao Yao, 2023) | 1.10 ± 0.00 | 1.86 ± 0.00 | 1.43 ± 0.00 | 0.050 ± 0.000 | 0.025 ± 0.000 | 0.027 ± 0.000 |
| | CLAP (Silva-Rodríguez et al., 2024) | 1.10 ± 0.00 | 1.83 ± 0.00 | 1.42 ± 0.00 | 0.047 ± 0.000 | 0.023 ± 0.000 | 0.026 ± 0.000 |
| ImageNet | ZS | 2.81 ± 0.00 | 10.07 ± 0.02 | 3.23 ± 0.00 | 0.086 ± 0.000 | 0.069 ± 0.000 | 0.084 ± 0.000 |
| | ZSLP | 2.48 ± 0.00 | 8.02 ± 0.02 | 2.88 ± 0.00 | 0.078 ± 0.000 | 0.062 ± 0.000 | 0.078 ± 0.000 |
| | CoOp (Zhou et al., 2022b) | 2.69 ± 0.00 | 7.33 ± 0.01 | 3.03 ± 0.00 | 0.081 ± 0.000 | 0.067 ± 0.000 | 0.081 ± 0.000 |
| | KgCoOp (Hantao Yao, 2023) | 2.65 ± 0.00 | 8.60 ± 0.02 | 3.01 ± 0.00 | 0.082 ± 0.000 | 0.066 ± 0.000 | 0.082 ± 0.000 |
| | CLAP (Silva-Rodríguez et al., 2024) | 2.44 ± 0.00 | 8.05 ± 0.01 | 2.82 ± 0.00 | 0.079 ± 0.000 | 0.062 ± 0.000 | 0.079 ± 0.000 |
| Oxford Pets | ZS | 1.05 ± 0.00 | 1.43 ± 0.00 | 1.33 ± 0.00 | 0.109 ± 0.000 | 0.067 ± 0.000 | 0.072 ± 0.000 |
| | ZSLP | 0.95 ± 0.00 | 1.26 ± 0.00 | 1.19 ± 0.00 | 0.080 ± 0.001 | 0.039 ± 0.000 | 0.038 ± 0.000 |
| | CoOp (Zhou et al., 2022b) | 0.95 ± 0.00 | 1.17 ± 0.00 | 1.13 ± 0.00 | 0.084 ± 0.000 | 0.036 ± 0.001 | 0.036 ± 0.001 |
| | KgCoOp (Hantao Yao, 2023) | 0.94 ± 0.00 | 1.20 ± 0.00 | 1.15 ± 0.00 | 0.079 ± 0.000 | 0.038 ± 0.001 | 0.037 ± 0.001 |
| | CLAP (Silva-Rodríguez et al., 2024) | 0.95 ± 0.00 | 1.28 ± 0.00 | 1.20 ± 0.00 | 0.084 ± 0.001 | 0.039 ± 0.000 | 0.038 ± 0.000 |
| Cars | ZS | 2.24 ± 0.00 | 3.09 ± 0.01 | 2.49 ± 0.00 | 0.109 ± 0.000 | 0.080 ± 0.000 | 0.102 ± 0.000 |
| | ZSLP | 1.62 ± 0.00 | 2.39 ± 0.01 | 1.94 ± 0.00 | 0.083 ± 0.000 | 0.061 ± 0.000 | 0.067 ± 0.000 |
| | CoOp (Zhou et al., 2022b) | 1.95 ± 0.00 | 2.64 ± 0.00 | 2.21 ± 0.00 | 0.087 ± 0.000 | 0.065 ± 0.000 | 0.078 ± 0.000 |
| | KgCoOp (Hantao Yao, 2023) | 1.99 ± 0.00 | 2.77 ± 0.00 | 2.29 ± 0.00 | 0.096 ± 0.000 | 0.069 ± 0.000 | 0.084 ± 0.000 |
| | CLAP (Silva-Rodríguez et al., 2024) | 1.63 ± 0.00 | 2.40 ± 0.00 | 1.96 ± 0.00 | 0.085 ± 0.000 | 0.061 ± 0.000 | 0.067 ± 0.000 |
| SUN397 | ZS | 2.83 ± 0.01 | 5.80 ± 0.01 | 3.21 ± 0.01 | 0.093 ± 0.000 | 0.073 ± 0.000 | 0.093 ± 0.000 |
| | ZSLP | 2.00 ± 0.00 | 3.96 ± 0.01 | 2.29 ± 0.00 | 0.080 ± 0.000 | 0.060 ± 0.000 | 0.072 ± 0.000 |
| | CoOp (Zhou et al., 2022b) | 2.35 ± 0.00 | 3.76 ± 0.01 | 2.64 ± 0.00 | 0.082 ± 0.000 | 0.068 ± 0.000 | 0.081 ± 0.000 |
| | KgCoOp (Hantao Yao, 2023) | 2.34 ± 0.00 | 4.18 ± 0.01 | 2.60 ± 0.00 | 0.083 ± 0.000 | 0.065 ± 0.000 | 0.082 ± 0.000 |
| | CLAP (Silva-Rodríguez et al., 2024) | 2.01 ± 0.00 | 3.99 ± 0.01 | 2.31 ± 0.00 | 0.080 ± 0.000 | 0.060 ± 0.000 | 0.072 ± 0.000 |
| UCF101 | ZS | 2.89 ± 0.01 | 5.24 ± 0.02 | 3.83 ± 0.01 | 0.124 ± 0.000 | 0.096 ± 0.000 | 0.124 ± 0.001 |
| | ZSLP | 1.58 ± 0.00 | 3.12 ± 0.01 | 1.98 ± 0.00 | 0.112 ± 0.001 | 0.074 ± 0.000 | 0.088 ± 0.000 |
| | CoOp (Zhou et al., 2022b) | 1.97 ± 0.01 | 3.12 ± 0.01 | 2.18 ± 0.01 | 0.114 ± 0.001 | 0.077 ± 0.000 | 0.110 ± 0.000 |
| | KgCoOp (Hantao Yao, 2023) | 1.74 ± 0.01 | 3.06 ± 0.01 | 2.06 ± 0.00 | 0.114 ± 0.001 | 0.070 ± 0.000 | 0.096 ± 0.000 |
| | CLAP (Silva-Rodríguez et al., 2024) | 1.63 ± 0.01 | 3.25 ± 0.01 | 2.00 ± 0.00 | 0.114 ± 0.001 | 0.076 ± 0.000 | 0.090 ± 0.000 |

Table 13: **Set size and CovGap** for CLIP (ViT-B) on 11 datasets across several few-shot adaptation approaches. Training for 16 shots.

| | | Set size (↓) | | | CovGap (↓) | | |
|---|---|---|---|---|---|---|---|
| | | LAC | APS | RAPS | LAC | APS | RAPS |
| Aircraft | ZS | 18.70 ± 0.06 | 19.75 ± 0.06 | 20.74 ± 0.05 | 0.138 ± 0.000 | 0.139 ± 0.000 | 0.127 ± 0.000 |
| | ZSLP | 8.14 ± 0.03 | 9.56 ± 0.03 | 9.43 ± 0.04 | 0.079 ± 0.001 | 0.069 ± 0.000 | 0.082 ± 0.001 |
| | CoOp (Zhou et al., 2022b) | 7.19 ± 0.02 | 8.09 ± 0.02 | 8.88 ± 0.04 | 0.080 ± 0.001 | 0.070 ± 0.000 | 0.084 ± 0.001 |
| | KgCoOp (Hantao Yao, 2023) | 7.53 ± 0.02 | 8.72 ± 0.03 | 9.13 ± 0.04 | 0.080 ± 0.001 | 0.075 ± 0.000 | 0.084 ± 0.001 |
| | CLAP (Silva-Rodríguez et al., 2024) | 8.20 ± 0.03 | 9.59 ± 0.03 | 9.43 ± 0.04 | 0.079 ± 0.001 | 0.070 ± 0.000 | 0.083 ± 0.001 |
| Caltech-101 | ZS | 0.94 ± 0.00 | 1.95 ± 0.01 | 1.20 ± 0.00 | 0.136 ± 0.001 | 0.092 ± 0.001 | 0.092 ± 0.001 |
| | ZSLP | 0.91 ± 0.00 | 1.12 ± 0.00 | 1.02 ± 0.00 | 0.151 ± 0.001 | 0.083 ± 0.001 | 0.082 ± 0.001 |
| | CoOp (Zhou et al., 2022b) | 0.91 ± 0.00 | 1.06 ± 0.00 | 1.02 ± 0.00 | 0.148 ± 0.001 | 0.084 ± 0.001 | 0.085 ± 0.001 |
| | KgCoOp (Hantao Yao, 2023) | 0.91 ± 0.00 | 1.19 ± 0.00 | 1.06 ± 0.00 | 0.150 ± 0.001 | 0.082 ± 0.001 | 0.083 ± 0.001 |
| | CLAP (Silva-Rodríguez et al., 2024) | 0.91 ± 0.00 | 1.22 ± 0.01 | 1.06 ± 0.00 | 0.146 ± 0.001 | 0.083 ± 0.001 | 0.085 ± 0.001 |
| DTD | ZS | 11.78 ± 0.06 | 13.30 ± 0.04 | 13.66 ± 0.06 | 0.122 ± 0.001 | 0.117 ± 0.001 | 0.121 ± 0.001 |
| | ZSLP | 2.91 ± 0.01 | 4.90 ± 0.03 | 3.34 ± 0.02 | 0.081 ± 0.001 | 0.071 ± 0.001 | 0.073 ± 0.001 |
| | CoOp (Zhou et al., 2022b) | 3.37 ± 0.02 | 4.50 ± 0.03 | 3.94 ± 0.04 | 0.084 ± 0.001 | 0.076 ± 0.001 | 0.081 ± 0.001 |
| | KgCoOp (Hantao Yao, 2023) | 3.53 ± 0.02 | 4.81 ± 0.03 | 4.04 ± 0.04 | 0.082 ± 0.001 | 0.073 ± 0.001 | 0.082 ± 0.001 |
| | CLAP (Silva-Rodríguez et al., 2024) | 3.03 ± 0.02 | 5.12 ± 0.03 | 3.48 ± 0.02 | 0.080 ± 0.001 | 0.069 ± 0.001 | 0.079 ± 0.001 |
| EuroSAT | ZS | 4.55 ± 0.01 | 4.78 ± 0.00 | 5.06 ± 0.01 | 0.103 ± 0.001 | 0.101 ± 0.001 | 0.102 ± 0.000 |
| | ZSLP | 1.46 ± 0.00 | 2.04 ± 0.00 | 1.84 ± 0.00 | 0.053 ± 0.000 | 0.045 ± 0.000 | 0.038 ± 0.000 |
| | CoOp (Zhou et al., 2022b) | 1.23 ± 0.00 | 1.59 ± 0.00 | 1.50 ± 0.00 | 0.041 ± 0.000 | 0.026 ± 0.000 | 0.031 ± 0.000 |
| | KgCoOp (Hantao Yao, 2023) | 1.28 ± 0.00 | 1.73 ± 0.00 | 1.61 ± 0.00 | 0.045 ± 0.000 | 0.032 ± 0.000 | 0.032 ± 0.000 |
| | CLAP (Silva-Rodríguez et al., 2024) | 1.53 ± 0.00 | 2.09 ± 0.00 | 1.90 ± 0.00 | 0.051 ± 0.000 | 0.044 ± 0.000 | 0.038 ± 0.000 |
| Flowers-102 | ZS | 9.26 ± 0.05 | 9.52 ± 0.06 | 15.15 ± 0.08 | 0.180 ± 0.000 | 0.175 ± 0.000 | 0.179 ± 0.000 |
| | ZSLP | 1.03 ± 0.00 | 2.19 ± 0.01 | 1.58 ± 0.00 | 0.132 ± 0.001 | 0.087 ± 0.001 | 0.091 ± 0.001 |
| | CoOp (Zhou et al., 2022b) | 0.93 ± 0.00 | 1.30 ± 0.00 | 1.17 ± 0.00 | 0.094 ± 0.001 | 0.078 ± 0.001 | 0.078 ± 0.001 |
| | KgCoOp (Hantao Yao, 2023) | 0.93 ± 0.00 | 1.44 ± 0.01 | 1.24 ± 0.00 | 0.102 ± 0.001 | 0.079 ± 0.000 | 0.078 ± 0.001 |
| | CLAP (Silva-Rodríguez et al., 2024) | 1.09 ± 0.00 | 2.30 ± 0.01 | 1.65 ± 0.00 | 0.135 ± 0.001 | 0.089 ± 0.001 | 0.094 ± 0.001 |
| Food-101 | ZS | 1.14 ± 0.00 | 1.89 ± 0.00 | 1.45 ± 0.00 | 0.054 ± 0.000 | 0.026 ± 0.000 | 0.031 ± 0.000 |
| | ZSLP | 1.08 ± 0.00 | 1.77 ± 0.00 | 1.39 ± 0.00 | 0.044 ± 0.000 | 0.023 ± 0.000 | 0.025 ± 0.000 |
| | CoOp (Zhou et al., 2022b) | 1.11 ± 0.00 | 1.64 ± 0.00 | 1.35 ± 0.00 | 0.050 ± 0.000 | 0.025 ± 0.000 | 0.030 ± 0.000 |
| | KgCoOp (Hantao Yao, 2023) | 1.08 ± 0.00 | 1.75 ± 0.00 | 1.38 ± 0.00 | 0.048 ± 0.000 | 0.022 ± 0.000 | 0.026 ± 0.000 |
| | CLAP (Silva-Rodríguez et al., 2024) | 1.07 ± 0.00 | 1.79 ± 0.00 | 1.39 ± 0.00 | 0.045 ± 0.000 | 0.023 ± 0.000 | 0.024 ± 0.000 |
| ImageNet | ZS | 2.81 ± 0.00 | 10.07 ± 0.02 | 3.23 ± 0.00 | 0.086 ± 0.000 | 0.069 ± 0.000 | 0.084 ± 0.000 |
| | ZSLP | 2.17 ± 0.00 | 6.60 ± 0.01 | 2.44 ± 0.00 | 0.076 ± 0.000 | 0.060 ± 0.000 | 0.074 ± 0.000 |
| | CoOp (Zhou et al., 2022b) | 2.39 ± 0.00 | 5.27 ± 0.01 | 2.73 ± 0.00 | 0.078 ± 0.000 | 0.064 ± 0.000 | 0.077 ± 0.000 |
| | KgCoOp (Hantao Yao, 2023) | 2.30 ± 0.00 | 5.95 ± 0.01 | 2.64 ± 0.00 | 0.078 ± 0.000 | 0.062 ± 0.000 | 0.077 ± 0.000 |
| | CLAP (Silva-Rodríguez et al., 2024) | 2.15 ± 0.00 | 6.85 ± 0.01 | 2.45 ± 0.00 | 0.077 ± 0.000 | 0.060 ± 0.000 | 0.074 ± 0.000 |
| Oxford Pets | ZS | 1.05 ± 0.00 | 1.43 ± 0.00 | 1.33 ± 0.00 | 0.109 ± 0.000 | 0.067 ± 0.000 | 0.072 ± 0.000 |
| | ZSLP | 0.94 ± 0.00 | 1.24 ± 0.00 | 1.17 ± 0.00 | 0.081 ± 0.001 | 0.040 ± 0.000 | 0.037 ± 0.000 |
| | CoOp (Zhou et al., 2022b) | 0.94 ± 0.00 | 1.16 ± 0.00 | 1.13 ± 0.00 | 0.083 ± 0.000 | 0.040 ± 0.000 | 0.041 ± 0.000 |
| | KgCoOp (Hantao Yao, 2023) | 0.94 ± 0.00 | 1.20 ± 0.00 | 1.15 ± 0.00 | 0.083 ± 0.001 | 0.038 ± 0.001 | 0.038 ± 0.000 |
| | CLAP (Silva-Rodríguez et al., 2024) | 0.95 ± 0.00 | 1.27 ± 0.00 | 1.19 ± 0.00 | 0.089 ± 0.001 | 0.039 ± 0.000 | 0.038 ± 0.001 |
| Cars | ZS | 2.24 ± 0.00 | 3.09 ± 0.01 | 2.49 ± 0.00 | 0.109 ± 0.000 | 0.080 ± 0.000 | 0.102 ± 0.000 |
| | ZSLP | 1.27 ± 0.00 | 1.98 ± 0.00 | 1.66 ± 0.00 | 0.082 ± 0.000 | 0.058 ± 0.000 | 0.057 ± 0.000 |
| | CoOp (Zhou et al., 2022b) | 1.37 ± 0.00 | 1.94 ± 0.00 | 1.66 ± 0.00 | 0.078 ± 0.000 | 0.057 ± 0.000 | 0.059 ± 0.000 |
| | KgCoOp (Hantao Yao, 2023) | 1.39 ± 0.00 | 2.04 ± 0.00 | 1.73 ± 0.00 | 0.080 ± 0.000 | 0.059 ± 0.000 | 0.061 ± 0.000 |
| | CLAP (Silva-Rodríguez et al., 2024) | 1.32 ± 0.00 | 2.04 ± 0.00 | 1.70 ± 0.00 | 0.080 ± 0.000 | 0.057 ± 0.000 | 0.057 ± 0.000 |
| SUN397 | ZS | 2.83 ± 0.01 | 5.80 ± 0.01 | 3.21 ± 0.01 | 0.093 ± 0.000 | 0.073 ± 0.000 | 0.093 ± 0.000 |
| | ZSLP | 1.72 ± 0.00 | 3.49 ± 0.01 | 2.09 ± 0.00 | 0.075 ± 0.000 | 0.055 ± 0.000 | 0.063 ± 0.000 |
| | CoOp (Zhou et al., 2022b) | 1.87 ± 0.00 | 2.73 ± 0.00 | 2.04 ± 0.00 | 0.073 ± 0.000 | 0.060 ± 0.000 | 0.072 ± 0.000 |
| | KgCoOp (Hantao Yao, 2023) | 1.85 ± 0.00 | 2.91 ± 0.01 | 2.07 ± 0.00 | 0.076 ± 0.000 | 0.059 ± 0.000 | 0.070 ± 0.000 |
| | CLAP (Silva-Rodríguez et al., 2024) | 1.76 ± 0.00 | 3.55 ± 0.01 | 2.12 ± 0.00 | 0.075 ± 0.000 | 0.056 ± 0.000 | 0.064 ± 0.000 |
| UCF101 | ZS | 2.89 ± 0.01 | 5.24 ± 0.02 | 3.83 ± 0.01 | 0.124 ± 0.000 | 0.096 ± 0.000 | 0.124 ± 0.001 |
| | ZSLP | 1.36 ± 0.00 | 2.85 ± 0.01 | 1.82 ± 0.00 | 0.106 ± 0.001 | 0.068 ± 0.000 | 0.077 ± 0.000 |
| | CoOp (Zhou et al., 2022b) | 1.51 ± 0.00 | 2.28 ± 0.01 | 1.76 ± 0.00 | 0.108 ± 0.001 | 0.068 ± 0.000 | 0.095 ± 0.000 |
| | KgCoOp (Hantao Yao, 2023) | 1.41 ± 0.00 | 2.47 ± 0.01 | 1.77 ± 0.00 | 0.106 ± 0.001 | 0.067 ± 0.000 | 0.081 ± 0.000 |
| | CLAP (Silva-Rodríguez et al., 2024) | 1.41 ± 0.00 | 2.94 ± 0.01 | 1.87 ± 0.00 | 0.108 ± 0.001 | 0.067 ± 0.001 | 0.079 ± 0.000 |