# OpenReview forum: "Are foundation models for computer vision good conformal predictors?"
_TMLR — Accepted by TMLR_

### Review · Reviewer_FNpJ · 2025-11-06

**Summary Of Contributions:**

This paper, from an empirical aspect, studies a very interesting problem: how would vision foundation models’ predictive performance correlate with their performance in (distribution-free) uncertainty quantification? Using metrics such as sizes of prediction sets, class-conditional coverage, and coverage gap, this paper offers a novel way to benchmark foundation models.

**Audience:**

Yes

**Audience Explanation:**

1. This paper conducts extensive numerical comparisons across various models and different choices of nonconformity scores, and also inspects the difference with/without distribution shifts and calibration. These experiments have enriched our knowledge on both vision foundation models and the practical performance of predictive inference.

2. While some experimental results are in line with our intuition, there are some experiments, especially those related to minimum class-conditional coverage and calibration, that are bringing in new findings in the field and can motivate more theoretical thinking.

3. This paper is clearly organized and experimental results are well presented.

**Claims And Evidence:**

No

**Claims Explanation:**

Overall, this paper conducts extensive numerical comparisons across various models and different choices of nonconformity scores, and also inspects the difference with/without distribution shifts and calibration. These experiments have enriched our knowledge on both vision foundation models and the practical performance of predictive inference.
But in some sections (e.g., Section 4.2.3 and discussions on page 7) are confusing in the sense that either the empirical results are hard to interpret or the intuition behind the results are not further theoretically justified by toy examples.

I have include strengths in the next part and also include my questions in "Requested Changes".

**Requested Changes:**

1. First and foremost, the notion of “better conformalization” is a bit vague even in this concrete context. In my understanding, it can relate to more accurate prediction sets, or more evenly distributed class-conditional performance, or better cross-domain robustness. I would suggest that the authors could formally and rigorously clarify this notion.

2. Regarding the metrics for evaluation, especially when combined with row 2 in Figure 1, the sign of the coverage gap would be very helpful in understanding the performance of different models. For example, it is unclear whether the increasing trend of the orange line in row 2-panel 1 in Figure 1 indicates a decreasing trend in coverage rate below 90% or otherwise.

3. On page 6, in the paragraph related to RAPS’s class-conditional coverage, I wonder if there is a typo in the definition of C_M(y).

4. The comparison between SSL models and models trained with supervised data on page 7 is a bit confusing. I also wonder how the performance on marginal coverage would be for two types of models?

5. Section 4.2.2 is very interesting in that it has shown very different performance of these nonconformity scores under distribution shift.
The robustness of APS may suggest that it can capture some domain-invariant information (or, this score is nearly independent of X). It would be more helpful if some discussions on the theoretical explanation could be added.

6. Section 4.2.3 on calibration offers a novel aspect of evaluating the performance of each model. However, the findings are very hard to interpret in the sense that, even with the “oracle” model (i.e. a model that captures the true distribution of Y|X), it may be not guaranteed to have the optimal performance in the class-conditional coverage. In this sense, the correlation between calibration and MCCC can be confusing without further theoretical justification (even with a toy setting).

---

> ### Author Response · Authors · 2025-11-28
> **Answer to reviewer FNpJ**
>
> We thank the reviewer for the detailed review, which we took into consideration in order to improve the quality of the paper. We have modified ambiguous terms, added the coverage in the corresponding Table, and clarified a few points. All updates are make in blue. Please find below the detailed answers.
>
> 1\. This is true, it could indeed indicate smaller set sizes, or lower coverage gap, or both. We have removed this term from the manuscript, and replaced it with more rigorous terms.
>
> 2\. This is an interesting question. Under the assumption of iid distributions between the conformal and test samples, conformal prediction ensures exact coverage. This is therefore the case in most of our experiments, including the one represented in Figure 1. In a perfect scenario, this coverage would be equal across all classes. However, in real applications, some classes show lower coverage, meaning that the coverage for other classes will be higher. Therefore, when there is an increase in the coverage gap, the absolute value of the gaps increases, but the average will remain at 0. If we misunderstood the concern, we would be happy to further discuss with the reviewer.
>
> 3\. Thank you for catching this! Indeed, the class-conditional coverage should be defined as $C_{M_i}(y)=\mathbb{P}(Y \in \mathcal{S}_{M_i}(X) \vert Y=y)$.
>
> 4\. We understand that this comparison may be confusing (please see the answer to question 4 from reviewer 2jeP). The marginal coverage is always guaranteed when there is no distribution shift between the calibration and test set (which is the case here). However, since this might cause confusion, we have added the coverage in this table (Table 1 in the paper).
>
> 5\. The difference in performance of APS and RAPS does not come from capturing domain-invariant information, as both are based on features extracted from the same model. Indeed, we can write $S_\text{RAPS} = S_\text{APS} + \lambda * (o(x,y)-k_{reg})^{+}$, meaning that RAPS has a regularization term, which penalizes larger prediction sets. This can be desired in a normal setting, but in a more realistic application where a distribution shift exists between the calibration and test sets, this term gives RAPS a strong incentive not to increase the set size, which comes at the cost of a strong decrease in coverage. We have added a sentence for better clarity on page 8.
>
> 6\. Thank you for raising this important point. We agree that the relationship between calibration and minimum class conditional coverage is not monotonic. As the reviewer correctly noted, even an oracle model (one that captures the true conditional distribution P(Y|X)) may not achieve optimal minimum class conditional coverage under conformal prediction. This stems from the fact that conformal methods (particularly those adaptive, such as APS or RAPS) guarantee marginal coverage, but class-conditional coverage depends on how nonconformity scores interact with the empirical conformal quantile across classes.
> Calibration reshapes the softmax distributions differently across classes, often reducing top-class confidence while increasing tail probabilities. This can inflate nonconformity scores, leading to larger prediction sets, which may improve MCCC at the cost of degrading efficiency. We note that the observed empirical correlation between calibration and MCCC is not universal, as it arises from the interplay between softened (or sharpened) score distributions and the different conformal mechanisms. We do not claim that calibration guarantees improved MCCC, but rather that it tends to improve empirically in our experiments, particularly for adaptive conformal methods.

---

### Review · Reviewer_iyJB · 2025-11-15

**Summary Of Contributions:**

The paper aims to conduct a comprehensive investigation of conformal inference applied to vision models for classification tasks. Several conformity scores and vision models are compared in the paper, based on which the authors provide guidelines for choosing between the scores and vision models.

**Additional Comments:**

N/A

**Audience:**

Yes

**Audience Explanation:**

It would be of interest to the practitioners to understand the performance of uncertainty quantification methods on vision data.

**Claims And Evidence:**

Yes

**Claims Explanation:**

The paper investigates the performance of 17 vision models with 3 conformity scores.

**Requested Changes:**

1. The paper only compares the classification task in the paper; apparently, the Vision Foundation model is capable of other tasks, e.g., generating new content. Providing a comparison in such tasks may strengthen the paper.

2. It would be informative to provide the SE of the quantities reported in the paper.

3. There are conformity scores addressing distribution shifts, e.g., [1,2]. They should also be included in the comparison.


[1] Tibshirani, Ryan J., et al. "Conformal prediction under covariate shift." Advances in neural information processing systems 32 (2019).

[2] Cauchois, Maxime, et al. "Robust validation: Confident predictions even when distributions shift." Journal of the American Statistical Association 119.548 (2024): 3033-3044.

---

> ### Author Response · Authors · 2025-11-28
> **Answer to reviewer iyJB**
>
> Thank you very much for your detailed review, which we took into consideration to improve our manuscript. We have updated the tables in order to add SE, updates are marked in blue in the manuscript. All updates are make in blue. Please find below the detailed answers.
>
> 1\. We thank the reviewer for their suggestion. We agree that vision foundation models are capable of a broad range of tasks, other than classification, including for example image generation. However the different nature of the predictions (e.g. probabilistic outputs in classification vs non probabilistic in image generation) makes the universal application of conformal prediction to a wide span of applications non trivial. For example, the non-conformity scores explored in this work cannot be used in non-probabilistic scenarios. While understanding the behaviour of conformal prediction across a wider range of applications could be interesting, we believe that this would dilute the message.
>
> 2\. This is indeed an interesting point, which could help to gain further insight. We have added them to the relevant tables in the supplementary material (Tables 5, 6, 7, 10, 11, 12, and 13).
>
> 3\. While there are some non-conformity scores addressing the issue of distribution shift, our main objective with this paper was to analyze how the most widely used and general non-conformity scores behave under a variety of scenarios, including, but not tailored to distribution shift. This is the reason why we chose these three non-conformity scores which enjoy widespread adoption across the community rather than specific scores that are designed for particular settings. Moreover, some of these distribution-shift-specific scores assume prior knowledge of the test set distribution, which is not aligned with our scenario.

---

> ### Comment · Action_Editor_Q35h · 2025-12-21
>
> Dear reviewer,
>
> Can you input your final recommendation?
>
> Best, AE

---

> ### Comment · Action_Editor_Q35h · 2025-12-25
>
> Dear reviewer,
>
> Can you input your final recommendation?
>
> Best, AE

---

### Review · Reviewer_2jeP · 2025-11-15

**Summary Of Contributions:**

Summary of contributions:
- Comprehensive and systematic study of conformal prediction on vision foundation models, covering 17 models and several representative datasets.
- Actionable insights from the comparison of CP methods, including the relationship between accuracy, coverage, and efficiency; performance under distribution shift; relationship of CP performance to pre-training strategies; impact of calibration on CP performance


Strengths:
- Evaluation is comprehensive
- The takeaway messages are practically useful
- The paper is well organized

Weakness
- Some data and model categorization information is missing
- Sometimes the writing can be a bit confusing

**Audience:**

Yes

**Audience Explanation:**

Researchers in computer vision and uncertainty quantification can build on the insights from this paper to build robust conformal prediction sets in their tasks. Also, people working on conformal prediction may be able to extract interesting theoretical and methodological questions based on these empirical findings.

**Claims And Evidence:**

Yes

**Claims Explanation:**

The conclusions and messages are from the empirical patterns, and the empirical studies cover a wide range of models, training strategies, and datasets, so the evidence is convincing.

**Requested Changes:**

1. Please pay attention to \cite{} and \citep{} throughout the paper.
2. In equations (4-6), what does $x_{k=y}$ mean? I suppose it is the predicted probability for label $y$, but the notation is a bit non-standard.
3. How is the data folds split in the experiments? Are the calibration and test data randomly split so they are exchangeable? If this is the case, then the coverage of conformal prediction is guaranteed to be above $1-\alpha$ on average, and the empirical coverage gap is mainly the artifact of randomness or ties. I was wondering whether coverage gap is a meaningful metric here.
4. Could you explain this sentence on page 7: "despite obtaining lower classification accuracy when using LP on the different foundation models, CP methods typically yield better performance than in ViTImageNet." I'm not sure I understand what you mean here. What are you comparing?
5. The message from the results surrounding Table 1 could be made more clear. Do you mean prediction accuracy does not necessarily translate to efficient prediction sets, and CLIP methods (or LP?) often leads to more efficient prediction sets?
6. Section 4.2.2 on distribution shifts is interesting. It might be interesting to further comment on why you choose ImageNet as calibration set and adapted version of ImageNet as test set, and not the opposite. Does this reflect practical deployment better?
7. In Table 3, there is a missing link between the training strategy category and the model names, so it is a bit confusing when mapping the table metrics to the conclusion remarks. Please consider add this information here and for other similar occurrences.
8. While APS has robust coverage, it seems its size is way larger than other methods. Would this be desired in practice?

---

> ### Author Response · Authors · 2025-11-28
> **Answer to reviewer 2jeP**
>
> Thank you very much for your detailed review, which really helped to improve our paper. We fixed the usage of citations, updated certain notations to improve clarity, moved metric definitions to the main paper, corrected an error in the description of the domain shift setting. All updates are make in blue. Please find below the detailed answers.
>
> 1\. We thank the reviewer for their comment regarding ```\cite{}``` and ```\citep{}``` and have updated the manuscript to ensure uniform usage throughout the revised version.
>
> 2\. Thank you for this comment. This notation was indeed not very standard; we have updated with $\pi_x(y)$ in Equations (4-6), along with a definition after Equation (4).
>
> 3\. **Data splits**. The data is first split into two sets. The first part is used to fit the linear probing head. The second is used for the conformal part, which we further split into a calibration and a test set. This split is performed randomly and repeated 100 times, with reported metrics averaged across runs. This procedure indeed ensures that the calibration and test sets are exchangeable and thus guarantees marginal coverage at $1-\alpha$, as expected under the conformal prediction framework (assuming no distribution shift is introduced).
>
> **Interpretation of coverage gap**. We believe there may be a misunderstanding regarding the interpretation of the coverage gap. This metric does not quantify the average deviation from the theoretical $1-\alpha$ across runs, but rather the average deviation between the nominal coverage $1-\alpha$ and the class conditional empirical coverage. In other words, even though marginal coverage is guaranteed by exchangeability, this does not imply that class-conditional coverage is uniformly close to $1-\alpha$.
> We acknowledge that placing the definition of coverage gap only in the appendix may have contributed to this confusion, and we propose moving it to the main paper for clarity.
>
> 4-5\. The objective of this experiment is to compare the conformal performance of models trained in a self-supervised versus a supervised manner. To ensure a fair scenario, we adopt the same architecture as the visual encoder for the different foundation models (i.e., ViT-B pre-trained on ImageNet). We can see that the accuracy of the model trained in a fully-supervised manner (i.e., by minimizing a cross-entropy loss between predictions and ground truth labels, defined as one-hot encoding vectors), when evaluated with a linear probing head is higher than those of models trained following a self-supervised paradigm. Nevertheless, despite the lower classification accuracy, the latter models yield more efficient (i.e., smaller prediction sets) and better conformal metrics (i.e., more uniform class-conditional coverage). These results indicate that the self-supervised nature of the models lead to better conformal performance. We added a sentence to clarify this conclusion : *This analysis should be put in perspective with the analysis in Figure 1, which found that higher accuracy leads to lower set size, indicating that the training scheme plays a very important role.*.
>
> 6\. We thank the reviewer for this comment. Unfortunately, we realized that a small error was made in this paragraph: the training of the model is performed on the original ImageNet and not on the variants. This does indeed better reflect deployment. In a real-world scenario, one would train the model and conformalize it on data from the same distribution. The more likely scenario is for the distribution shift to occur when the model is deployed (i.e., between the training/conformal set and the test set). This is the case in this experiment. We have updated the manuscript accordingly.
>
> 7\. This was indeed unclear. We have updated the manuscript to make the link between the names in Table 3 and the paragraph more explicit. Please correct us if we have misunderstood your question.
>
> 8\. We agree that the question of when to require more robust coverage, at the cost of a larger set size, is very important. We had addressed this question in the conclusion. The choice depends primarily on the sensitivity of the application. When minimizing certain types of errors is crucial, even at the cost of bigger conformal sets, APS may be preferable. This is particularly relevant in medical applications, where practitioners can afford to spend additional time analyzing challenging cases.

---

> > ### Comment · Action_Editor_Q35h · 2025-12-25
> >
> > Dear reviewer,
> >
> > Can you input your final recommendation?
> >
> > Best, AE

---

> ### Comment · Action_Editor_Q35h · 2025-12-21
>
> Dear reviewer,
>
> Can you input your final recommendation?
>
> Best, AE

---

### Decision · Action_Editor_Q35h · 2025-12-26

**Recommendation:** Accept as is

**Audience:**

Yes

**Audience Explanation:**

Researchers in computer vision and uncertainty quantification can use the insights from this paper to build robust conformal prediction sets in their tasks.

**Claims And Evidence:**

Yes

**Claims Explanation:**

From an empirical perspective, this paper investigates a compelling question: how does the predictive performance of vision foundation models correlate with their ability to quantify uncertainty in a distribution-free manner? By utilizing metrics such as prediction set size, class-conditional coverage, and coverage gap, the authors introduce a novel framework for benchmarking these models.

Overall, the study presents comparisons across a variety of models and nonconformity scores. It also examines performance differences both with and without distribution shifts and calibration. These experiments deepen our understanding of vision foundation models and the practical application of predictive inference.

After receiving the first-round reviews, the authors made improvements on the manuscript and all three reviewers recommended acceptance of the paper.